# Mapping the mouse Allelome reveals tissue-specific regulation of allelic expression

Daniel Andergassen[1†], Christoph P Dotter[1‡], Daniel Wenzel[2], Verena Sigl[2], Philipp C Bammer[1§], Markus Muckenhuber[1¶], Daniela Mayer[1§], Tomasz M Kulinski[1**], Hans-Christian Theussl[3], Josef M Penninger[2], Christoph Bock[1], Denise P Barlow[1*], Florian M Pauler[1*‡], Quanah J Hudson[1*††]

[1]CeMM, Research Center for Molecular Medicine of the Austrian Academy of Sciences, Vienna, Austria; [2]IMBA, Institute of Molecular Biotechnology of the Austrian Academy of Sciences, Vienna, Austria; [3]IMP/IMBA Transgenic Service, Institute of Molecular Pathology, Vienna, Austria

*For correspondence:
dpbarlow65@gmail.com (DPB);
florian.pauler@ist.ac.at (FMP);
quanah.hudson@imba.oeaw.ac.at
(QJH)

Present address: [†]Department of Stem Cell and Regenerative Biology, Harvard University, Cambridge, United States; [‡]Institute of Science and Technology Austria, Klosterneuburg, Austria; [§]Friedrich Miescher Institute for Biomedical Research, Basel, Switzerland; [¶]German Cancer Research Center, Heidelberg, Germany; [**]Institute of Biochemistry and Biophysics Polish Academy of Sciences, Warszawa, Poland; [††]Institute of Molecular Biotechnology of the Austrian Academy of Sciences, Vienna, Austria

Competing interests: The authors declare that no competing interests exist.

**Abstract** To determine the dynamics of allelic-specific expression during mouse development, we analyzed RNA-seq data from 23 F1 tissues from different developmental stages, including 19 female tissues allowing X chromosome inactivation (XCI) escapers to also be detected. We demonstrate that allelic expression arising from genetic or epigenetic differences is highly tissue-specific. We find that tissue-specific strain-biased gene expression may be regulated by tissue-specific enhancers or by post-transcriptional differences in stability between the alleles. We also find that escape from X-inactivation is tissue-specific, with leg muscle showing an unexpectedly high rate of XCI escapers. By surveying a range of tissues during development, and performing extensive validation, we are able to provide a high confidence list of mouse imprinted genes including 18 novel genes. This shows that cluster size varies dynamically during development and can be substantially larger than previously thought, with the *Igf2r* cluster extending over 10 Mb in placenta.

## Introduction

Allele-specific expression can occur in different contexts during mammalian development and affect a wide-range of processes. Random monoallelic expression at the single-cell level has been reported to be relatively common and plays an important role in the maturation of the lymphoid cell lineage where allelic exclusion of T and B cell receptors is required (*Reinius and Sandberg, 2015*). At the tissue level such cases appear biallelic, but genetic and epigenetic differences between the alleles can lead to allele-specific biases in populations of cells or the whole organism.

Genetic differences between the alleles of mammalian genes frequently cause allele-specific expression differences in human and mouse (*Geuvadis Consortium et al., 2013*; *Crowley et al., 2015*). The sequence of the two alleles can vary at single-nucleotide polymorphisms (SNPs) that can alter gene expression by modulating transcription factor binding to gene promoters or distal and proximal activating regions called enhancers (*Leung et al., 2015*). Active enhancers are marked by the H3K27ac histone modification (*Creyghton et al., 2010*) and can activate one or more promoters by direct interaction (*Shlyueva et al., 2014*). Allelic expression can also be caused by epigenetic differences between the alleles, notably in the developmentally important processes of X chromosome inactivation (XCI) and genomic imprinting. In both cases, a long non-coding (lnc) RNA can cause the

initiation of silencing, with *Xist* initiating XCI, and some clusters of imprinted genes being silenced by an imprinted lncRNA.

In female mammals, the *Xist* lncRNA is expressed from one of the two X chromosomes leading to widespread epigenetic silencing of X-linked genes apart from a subset that escape XCI, reported to be 3% in mouse and 15% in human (*Berletch et al., 2011*), although other reports indicate that the number of escapers in mouse may be higher at around 13% (*Calabrese et al., 2012*). *Xist* uses the three-dimensional structure of the X-chromosome to gain access to distant parts of the chromosome from which it spreads to eventually coat the whole inactive X and cause XCI (*Engreitz et al., 2013*). Current evidence indicates that *Xist* initiates silencing by interacting with SPEN that then recruits HDAC3 to cause hypoacetylation of the X chromosome (*Chu et al., 2015*; *McHugh et al., 2015*; *Monfort et al., 2015*). A series of factors are then recruited that establish the repressive chromatin state required to maintaining silencing, including the Polycomb Repressive Complexes 1 and 2 (PRC1 and 2), DNMT1, SAF-A and ASH2L (*Wutz, 2011*).

Imprinted genes are mostly clustered with allele-specific silencing regulated by a distant differential DNA methylated imprint control element (ICE). In the most common mechanism, the unmethylated ICE acts as a promoter for a lncRNA that silences a cluster of genes, as shown for *Airn* and *Kcnq1ot1* in the *Igf2r* and *Kcnq1* clusters (*Mancini-Dinardo et al., 2006*; *Sleutels et al., 2002*). Both *Airn* and *Kcnq1ot1* have been associated with the histone-modifying enzymes EHMT2 and PRC2, and are thought to guide deposition of H3K9me2 and H3K27me3 to silence distant genes in these clusters (*Nagano et al., 2008*; *Terranova et al., 2008*). However, *Airn* directly silences the overlapped *Igf2r* by transcriptional interference, a process not requiring these enzymes (*Latos et al., 2012*; *Mager et al., 2003*; *Nagano et al., 2008*). It has been hypothesized that disruption of enhancer activity may be the first step in initiating silencing of imprinted genes distant from the lncRNA locus (*Pauler et al., 2012*).

Extensive studies on the influence of SNPs on allelic expression and disease association have been performed in human adult tissues or cell culture models (*Leung et al., 2015*). RNA-seq on mouse tissues from F1 crosses have been used to detect expression quantitative trait loci (eQTLs) (*Keane et al., 2011*; *Lagarrigue et al., 2013*), escape from XCI (*Berletch et al., 2015*) and imprinted expression (*DeVeale et al., 2012*; *Babak et al., 2008*, *2015*; *Okae et al., 2012*; *Wang et al., 2011*, *2008*), but studies of total allelic expression have been lacking. We have pioneered an approach to classify allelic expression of all genes in a tissue from RNA-seq data (*Andergassen et al., 2015*), and apply this here to map the allelic expression states of protein-coding (pc) and non-coding (nc) genes in 23 different mouse tissues and developmental stages to define the mouse Allelome. This revealed that biases in allelic expression of pc-genes are highly tissue-specific in agreement with previous reports (*Babak et al., 2015*; *Prickett and Oakey, 2012*), while nc-genes tended to show a consistent bias when expressed. Following this, in the 19 females tissues we confirmed reports that XCI escapers can be tissue-specific (*Berletch et al., 2015*), and found an unusually high proportion of escapers in leg muscle (>50%). By assembling a high confidence list of validated or supported imprinted genes, we found that an even larger proportion than previously thought belong to clusters (>90%), that these clusters can be much larger than previously reported, and that they expand and contract during development, reaching their maximum in extra-embryonic tissues. In particular, we found that the *Igf2r* cluster expanded to 10 Mb in placenta, representing the largest *cis* co-regulated region outside of the X chromosome. For all types of allelic expression that we investigated we found an association with nearby allele-specific H3K27ac enrichment, indicating that allele-specific expression may be mediated through genetic differences in enhancers or by epigenetic repression established on enhancers during XCI and imprinted silencing.

## Results

### The mouse gene expression Allelome shows tissue-specific variation

To investigate how allelic expression varies between tissues and during development, we first generated a near complete picture of allelic expression, or the mouse Allelome. We chose a range of 23 pluripotent, embryonic, extra-embryonic, neonatal and adult tissues, including a developmental series for selected tissues (*Figure 1A*, *Supplementary file 1*, sheets A-B). We placed an emphasis on tissues where imprinted expression has been suggested to play an important role, such as in the

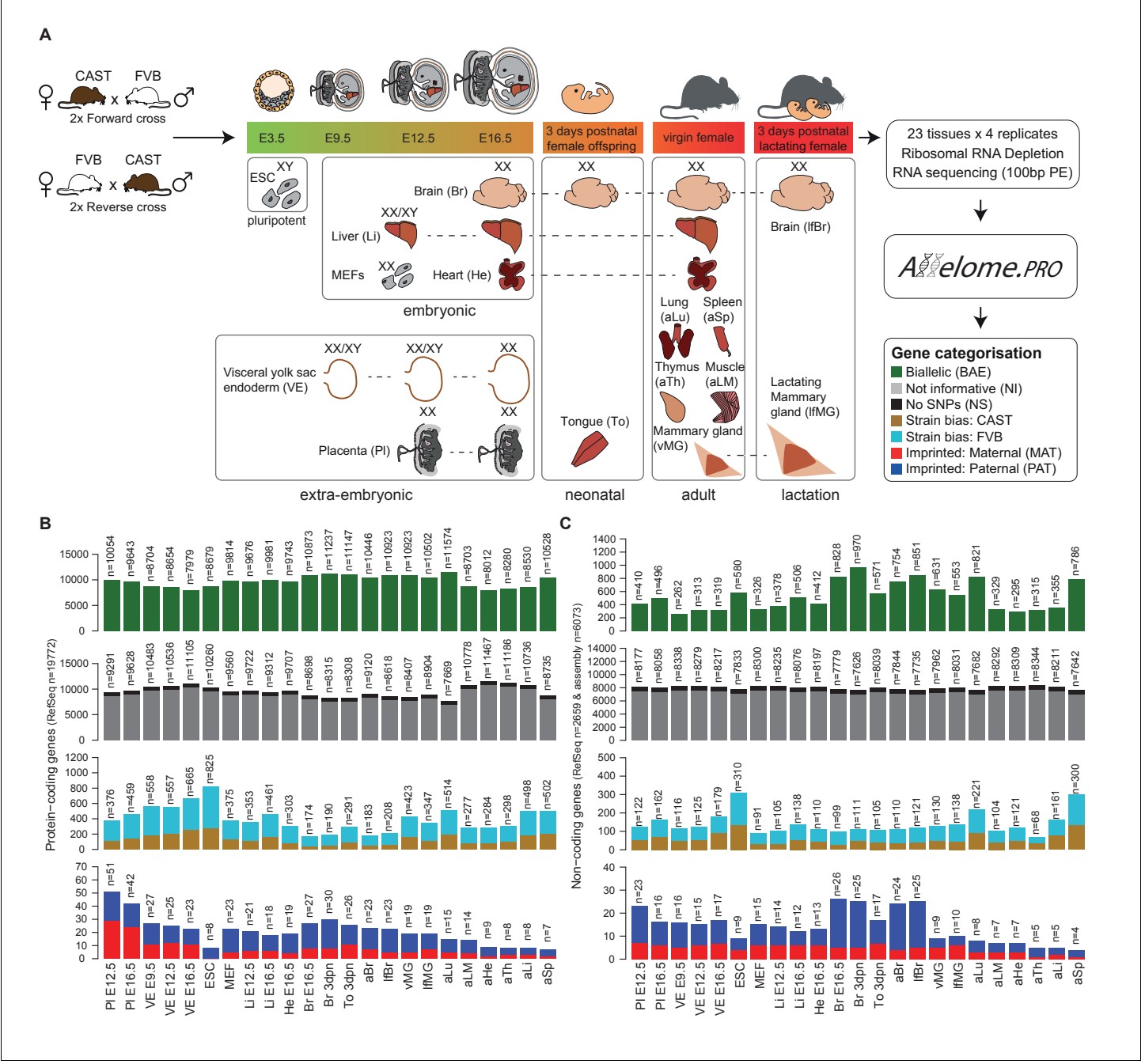

**Figure 1.** Defining the mouse Allelome. (A) Strategy for detecting allelic expression from RNA-seq data from 23 mouse tissues and developmental stages using Allelome.PRO. Every gene in the annotation is classified into one of seven different allelic expression categories indicated by different colors in the key and explained in the text. These colors are used in figures throughout the manuscript. The sex of the tissues is indicated by XX (female) and XY (male). Individuals were used except for indicated embryonic tissues where an entire litter was pooled (XX/XY). (B) Allelome.PRO classification of the allelic expression status of protein-coding genes in each tissue. (C) Allelome.PRO classification of non-coding genes. Tissues examined were placenta (Pl embryonic day (E) 12.5, E16.5), visceral yolk sac endoderm (VE E9.5, E12.5, E16.5), embryonic stem cells (ESC), mouse embryonic fibroblasts (MEF E12.5), embryonic liver (Li E12.5, E16.5), embryonic heart (He E16.5), embryonic and neonatal brain (Br E16.5, 3 days postnatal (dpn)), neonatal tongue (To 3dpn), adult brain (aBr), adult lactating female brain (lfBr), adult virgin mammary glands (vMG), adult lactating female mammary glands (lfMG), adult lung (aLu), adult leg muscle (aLM), adult heart (aHe), adult thymus (aTh), adult liver (aLi) and adult spleen (aSp). Embryo and placenta diagrams adapted from *Hudson et al. (2011)*. Allelome.PRO settings: FDR 1%, allelic-ratio cutoff 0.7, minread 2.

The following figure supplements are available for figure 1:

*Figure 1 continued on next page*

*Figure 1 continued*

**Figure supplement 1.** Clustering of tissues by their RNA-seq expression data matches the expected developmental relationships.
**Figure supplement 2.** The Allelome.PRO pipeline output and quality controls.
**Figure supplement 3.** Novel non-coding RNAs show a similar non-coding potential to annotated Refseq non-coding RNAs.

energy transfer between the mother and embryo and in neonatal and maternal behavior (*Peters, 2014*; *Stringer et al., 2014*), which includes tissues like brain and placenta reported to show the most imprinted expression (*Babak et al., 2015*). Therefore, our samples include a developmental series of brain and the extra-embryonic placenta and visceral yolk sac endoderm (VE) tissues, as well as the neonatal tongue and virgin and lactating mammary gland and brain from the lactating female.

For each tissue, we collected four F1 samples from two reciprocal crosses between FVB/NJ (FVB) and CAST/EiJ (CAST) mice. To enable analysis of X chromosome allelic expression, we collected single female (XX) organs, except for embryonic day (E) 12.5 liver, E9.5 VE and E12.5 VE where tissues from a litter were pooled (mix of XX/XY), and embryonic stem (ES) cells, which were derived from male (XY) blastocysts. We performed total RNA sequencing (RNA-seq) to maximise sensitivity by detecting SNPs in the introns, and also to allow detection of non-polyadenylated nc-genes. Unsupervised clustering confirmed the quality of the dataset by showing that replicates of the same tissue clustered together, closest to the same organ at different developmental stages as expected (*Figure 1—figure supplement 1*). We analyzed these data for biases in allelic expression using Allelome.PRO with a false discovery rate (FDR) of 1% (based on mock comparisons of the allelic score (the $\log_{10}(p)$ value for deviations from an allelic ratio of 0.5 calculated based on the binomial distribution)), an allelic ratio cutoff of 0.7 (*Andergassen et al., 2015*), a custom annotation, and SNPs from the Sanger database (*Keane et al., 2011*). We previously validated the Allelome.PRO strategy in F1 crosses of inbred mouse strains (*Andergassen et al., 2015*), and the approach is described in more detail in this paper and accompanying manual, as well as in the Materials and methods and *Figure 1—figure supplement 2A*. The pipeline requires a single allelic ratio cutoff in order to be able to classify all annotated genes into allelic expression categories. We chose an allelic ratio cutoff of 0.7, because previous analysis showed that most known imprinted genes were above this level, as were known strain biased genes on the X chromosome due to a bias in X chromosome inactivation in CAST/FVB F1 mice (*Andergassen et al., 2015*). To generate a comprehensive annotation that covered all transcripts present in our dataset, we combined the RefSeq mouse annotation for pc- and nc-genes (*Pruitt et al., 2014*), with nc-loci not in RefSeq detected by reference based assembly from our data, as detailed in the Materials and methods. Analysis with RNAcode and CPC indicated that the coding potential of our nc-loci was significantly less than for pc-genes, but not distinguishable from RefSeq nc-genes (*Kong et al., 2007*; *Washietl et al., 2011*) (*Figure 1—figure supplement 3*). In summary, our combined annotation had a total of 20743 pc-gene and 9068 nc-gene loci (including 2778 RefSeq nc-genes).

Using this approach, we classified allelic expression of pc- and nc-genes in the above 23 tissues as showing biallelic expression (BAE), not informative due to no or low expression (NI), not informative due to no SNPs (NS), strain-biased toward the CAST or FVB allele, imprinted maternal expression (MAT) or paternal expression (PAT) (*Figure 1*, *Figure 2—figure supplement 1A*). Biallelic genes included genes with a consistent strain bias where the median of the replicates is below the allelic ratio cutoff (40.4–57.0%), genes that fluctuate in the direction of strain bias between the replicates below the allelic ratio cutoff (36.7–50.1%), and genes that fluctuate in the direction of strain bias with at least one replicate above the allelic ratio cutoff (4.8–10.4%) (*Figure 1—figure supplement 2B*). For autosomes, the total number of BAE pc-genes showed limited variation between tissues and developmental stages, varying 1.4-fold between 7979–11,574 genes from the 19,772 annotated pc-genes, or 40–59% of the total (*Figure 1B*, first row). The number of non-informative pc-genes showed a reciprocal pattern for each tissue, varying between 7669–11467 genes (39–58.0% of the total), while 723 genes (3.7%) could not be assessed due to a lack of SNPs (*Figure 1B*, second row). Low tissue-specific variation in the number of BAE pc-genes is partly explained by

genes that showed biallelic expression in multiple tissues, with 31% of biallelic genes showing biallelic expression in all 23 tissues (*Figure 2—figure supplement 1B*, top). In contrast, the number of pc-genes showing strain-biased and imprinted expression varied greatly between tissues. Genes showing strain-biased expression varied 4.7-fold among the different tissues from between 174–825 genes, or 0.9–4.2% of the total (*Figure 1B*, third row). Overall strain bias toward the FVB allele was 1.9-fold higher than strain bias toward the CAST allele, which may reflect an alignment bias due to FVB having a shorter genetic distance to the C57BL/6 reference genome. Genes showing imprinted expression showed the most tissue-specific variation, varying 7.2-fold between the different tissues from 7 to 51 genes, or 0.035–0.258% of the total (*Figure 1B*, fourth row). Overall, there was no difference in the distribution of allelic ratio of autosomal genes between the tissues with all tissues showing a median allelic ratio around 0.5 (*Figure 1—figure supplement 2C*).

The proportion of autosomal nc-genes classified BAE per tissue was much lower than for pc-genes, and showed greater tissue-specific variation, varying 3.7-fold from 262 to 970 genes or 3.0–11.1% of the total (*Figure 1C*, first row). High variation is likely due to the known tight tissue-specific expression of lncRNAs that make up most of the nc-genes (*Necsulea et al., 2014*), as was further indicated by the high proportion of nc-genes that were non-informative in each tissue (87.3–95.5% of the total, *Figure 1C*, second row). Reflecting this, in contrast to pc-genes only a small minority of nc-genes were biallelically expressed in all 23 tissues (50 of 2673, 1.8%) and the majority showed biallelic expression in one tissue only (963 of 2673, 36.9%) (*Figure 2—figure supplement 1B*, bottom). As for pc-genes, a low proportion of nc-genes could not be assessed due to a lack of SNPs (700 genes or 8.0% of the total). There was a similar high degree of variation in the number of nc-genes showing strain bias (4.5 fold, 68–310 transcripts, 0.77–3.5% of the total) or imprinted expression (6.5 fold, 4–26 transcripts, 0.045–0.29% of the total) as was seen for pc-genes (*Figure 1C*, third and fourth rows).

In summary, mapping the Allelome revealed tissue-specific variation in the number of strain-biased and imprinted genes for both pc- and nc-genes, while the number of BAE genes was similar between tissues for pc-, but not nc-genes. Interestingly, the number of pc- and nc-genes in each allelic expression category appeared to co-vary between tissues, with the total number of pc- and nc-informative genes showing a high correlation ($r^2 = 0.77$, $p<10^{-4}$, Pearson).

## Tissue-specific strain-biased expression correlates with strain-biased enhancer marks

The variation in the absolute number of strain-biased pc- and nc-genes was also reflected in the proportion of strain-biased genes relative to the number of informative genes per tissue (*Figure 2A*). The proportion of pc-genes showing strain-biased expression (1.6% (embryonic brain) - 8.7% (ESC)) was generally lower than for nc-genes (10.0% (neonatal brain) - 34.8% (E16.5 VE)). This may reflect the known high evolution rate of lncRNAs that may lead to strain-specific lncRNAs (*Necsulea et al., 2014*) and is in line with recent findings that lncRNAs vary significantly more than pc-genes between people (*Kornienko et al., 2016*). However, we found that the number of pc- and nc- strain-biased genes detected per tissue was correlated ($r^2 = 0.71$, $p<10^{-3}$, Pearson), indicating that some may be co-regulated.

We next investigated if the allelic status of genes is constant between tissues. We found that most biallelic genes remained biallelic wherever they are expressed for both pc- and nc-genes (*Figure 2B* left), although the majority of nc-genes were expressed only in one tissue, whereas most pc- genes are expressed in multiple tissues (*Figure 2—figure supplement 1B*). Most strain-biased nc-genes did not change their allelic status between tissues, whereas pc-genes could be categorized into two groups based on whether they maintained their allelic status between tissues or not (*Figure 2B* right). The first group (134 CAST and 249 FVB) maintained strain-biased expression in 95–100% of the tissues where they were expressed, whereas the second group containing the majority of strain-biased genes (433 CAST and 569 FVB) maintained their allelic status in only 5–90% of the tissues. Further indicating that strain-biased expression is tissue-specific and that genes can switch their allelic expression status between tissues, unsupervised clustering using the allelic ratio of strain-biased genes that were informative in all tissues largely reflected the developmental relationship of the tissues (*Figure 2—figure supplement 1C*).

We next sought to determine if tissue-specific strain-biased expression may be explained by the activity of strain-biased enhancers. To investigate this we used Allelome.PRO to detect allelic

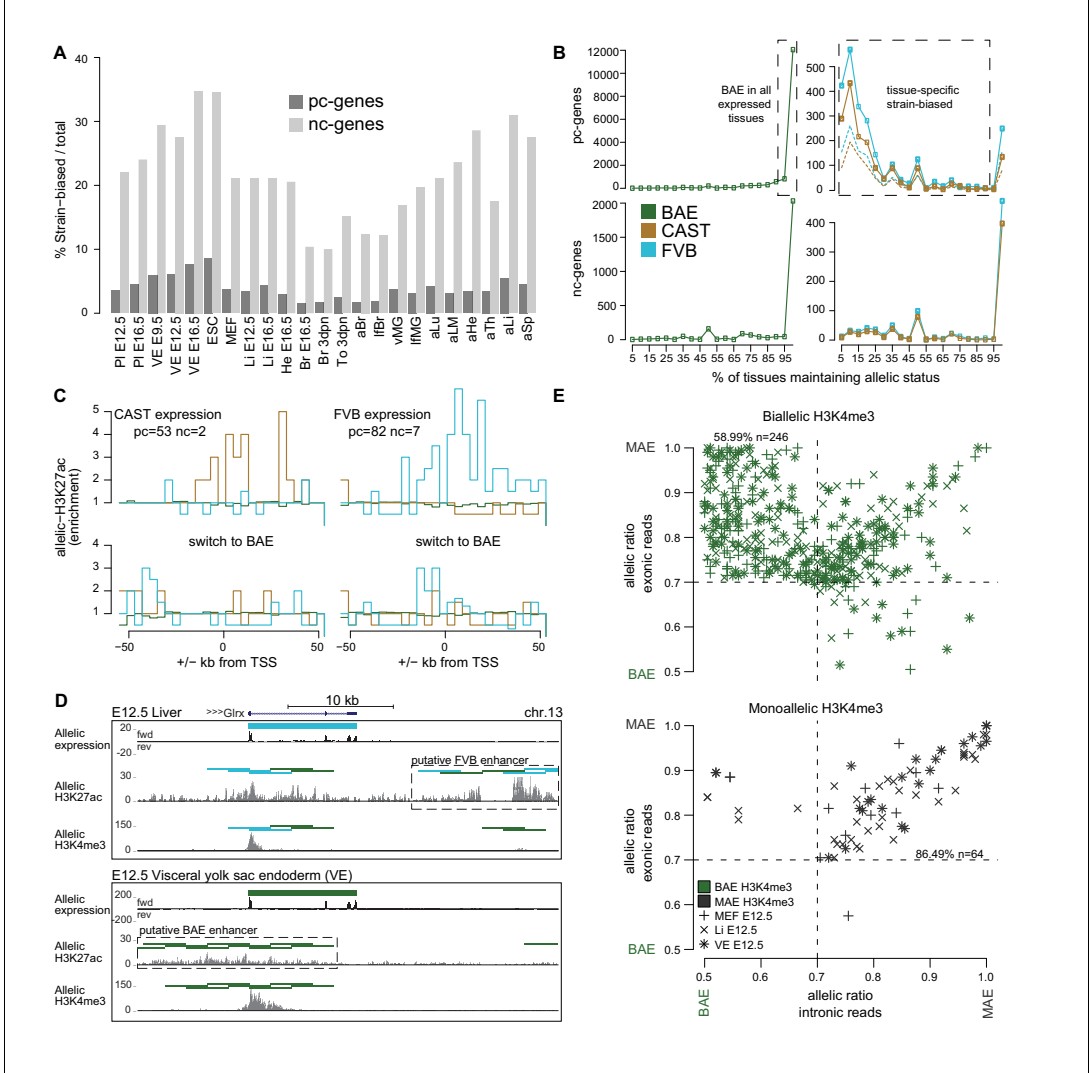

**Figure 2.** The Allelome reveals tissue-specific expression of strain-biased genes. (**A**) The percentage of strain-biased genes from total informative genes for each tissue for protein-coding (pc, black) and non-coding (nc, grey) genes. (**B**) The percentage of tissues where pc- and nc- genes maintained their biallic (left) or strain-biased (right), allelic expression status (calculated relative to number of tissues where a gene was informative). Allelome.PRO settings: FDR 1%, allelic-ratio cutoff 0.7, minread 2, dotted lines indicates the outcome with a 0.8 allelic ratio cutoff. (**C**) The enrichment of H3K27ac ±50 kb from the transcription start site (TSS) of genes that show strain-biased expression in either E12.5 VE or Li, and biallelic expression in the other tissue. Top: H3K27ac enrichment near strain-biased genes. The enrichment over random of allelic H3K27ac 4 kb windows was calculated. Bottom: The same analysis for the same set of genes where they show biallelic expression. Analysis detailed in Materials and methods. (**D**) An example of putative enhancer switching: *Glrx* switches from FVB strain-biased expression in liver to BAE expression in VE. This is associated with a switch in putative enhancers that matches the allelic expression status. Allelome.PRO settings for H3K27ac: FDR 1%, allelic-ratio cutoff 0.7, minread 1. (**E**) Strain-biased genes with biallelic H3K4me3 on their promoter are enriched for genes that show biallelic expression in their introns and strain-biased expression in their exons, indicating post-transcriptional differences in stability may explain strain-biased expression. Scatter plots comparing the allelic ratio of strain-biased genes in their exons and introns. Top: Strain-biased genes with biallelic H3K4me3 ChIP-seq enrichment on their promoter. Bottom: Strain-biased genes with supporting monoallelic enrichment of H3K4me3 on their promoter (key indicates H3K4me3 color code and tissue). Color code as in *Figure 1*.

The following figure supplement is available for figure 2:

**Figure supplement 1.** Characterization of the mouse Allelome reveals that protein-coding genes switch their allelic status among tissue and development.

enrichment of H3K27ac chromatin immunoprecipitation and sequencing (ChIP-seq) data from FVB x CAST reciprocal crosses for fetal liver and VE, and compared this to the RNA-seq analysis for these tissues. In both fetal liver and VE, we found that H3K27ac informative windows were mostly biallelic, with only between 1.6% (VE) and 11.3% (fetal liver) informative windows showing a strain bias (*Figure 2—figure supplement 1D*). We chose genes that switched from strain-biased expression in one tissue to biallelic expression in the other, and then examined H3K27ac enrichment ±50 kb from the transcription start site (TSS) (detailed in the Materials and methods). For both CAST and FVB strain-biased genes, we found strain-biased H3K27ac enrichment both upstream and downstream of the TSS matching the strain-biased expression, while no enrichment was seen when the same genes were biallelic in the other tissue (*Figure 2C*, *Supplementary file 1*, sheet C, note that BAE enrichment was not expected since BAE windows were overrepresented in the genome [*Figure 2—figure supplement 1D*]). This enrichment was explained by 39/144 (27%) strain-biased to BAE switchers (*Supplementary file 1*, sheet C), such as *Glrx* where a change from FVB biased expression in liver to BAE in VE was correlated with a putative switch in enhancer usage matching the allelic expression status (*Figure 2D*). To investigate if other strain-biased expression may be explained by allele-specific differences in post-transcriptional degradation, we took the strain-biased genes for E12.5 MEFs, fetal liver and VE where we also had matching H3K4me3 data, and calculated the allelic ratio separately for the introns (nascent transcript) and exons (mature transcript). The set of strain-biased genes with biallelic H3K4me3 over their promoters were enriched for genes with biallelic introns (nascent transcript) but strain-biased exons (mature transcript), indicating that these genes may be strain-biased due to post-transcriptional differences in stability (59%, *Figure 2E*). In contrast, strain-biased genes with monoallelic H3K4me3 on their promoters were strongly enriched for genes that showed a strain bias in both the introns and exons (86%, *Figure 2E*), indicating that the bias occurred at the transcriptional level presumably due to SNPs in regulatory regions such as enhancers as indicated previously (*Figure 2C,D*). Altogether these results indicate that tissue-specific strain-biased expression may occur due to a switch in enhancer usage between tissues, from an enhancer that shows strain-biased activity to one that shows biallelic activity, or due to tissue-specific differences in post-transcriptional stability between the alleles.

## Escape from X-inactivation is tissue-specific and correlates with increased distance from monoallelic enhancers

We used 19 female tissues (*Figure 1A*) to define the Allelome for the X-chromosome, 16 epiblast-derived embryonic, neonatal and adult tissues showing random X chromosome inactivation (XCI), and three extra-embryonic tissues showing imprinted XCI (*Figure 3A*, *Figure 3—figure supplement 1A*). In inbred mouse strains, both X-chromosomes in epiblast-derived tissues have an equal chance to express the *Xist* lncRNA gene leading to random inactivation; however, in CAST/FVB F1 mice the FVB allele is preferentially inactivated due to a bias in *Xist* expression (*Calaway et al., 2013*; *Chadwick et al., 2006*). In extra-embryonic tissues *Xist* is expressed only from the paternal allele, leading to inactivation of the paternal allele (*Kay et al., 1994*). Therefore, in this F1 cross XCI escapers can be detected as genes that do not show the expected CAST bias (around 0.7 allelic ratio in epiblast-derived tissues) or MAT bias (close to an allelic ratio of 1.0 in extra-embryonic tissues) in expression, but rather show BAE or an unexpected bias (note: to conservatively call BAE escapers a lower allelic ratio cutoff of 0.6 was used for this analysis, see Materials and methods for details). Using this approach across the 19 XX female tissues, we detected 250 candidate escaper genes (225 random XCI and 43 imprinted XCI escapers, 18 escape in both) from 1044 X chromosome genes (792 pc- and 252 nc-genes), with a further 178 genes (14.6%) unable to be assessed due to a lack of SNPs (*Figure 3A*, *Supplementary file 1*, sheets D-E). These included 31 out of 55 previously reported XCI escaper genes (*Berletch et al., 2015*; *Finn et al., 2014*; *Wu et al., 2014*). The high number of escapers detected could be due to the sensitivity of our method, as there was a tendency for more escapers to be detected at lower expression levels, but when the number of escapers at different expression levels was normalized for the number of informative genes, then there was no difference indicating this was not an artifact (*Figure 3—figure supplement 1B*). The number of genes escaping XCI varied considerably between tissues from 1 to 108 pc-genes (0.4–52.1% of informative pc-genes), and from 1 to 10 for nc-genes including *Xist* (0.4–71.4% of informative nc-genes) (*Figure 3B* top, middle). Genes that escaped in a high proportion of tissues, or ubiquitous escapers, included a high number of known escapers such as *Kdm6a* and the nc-gene *Firre*

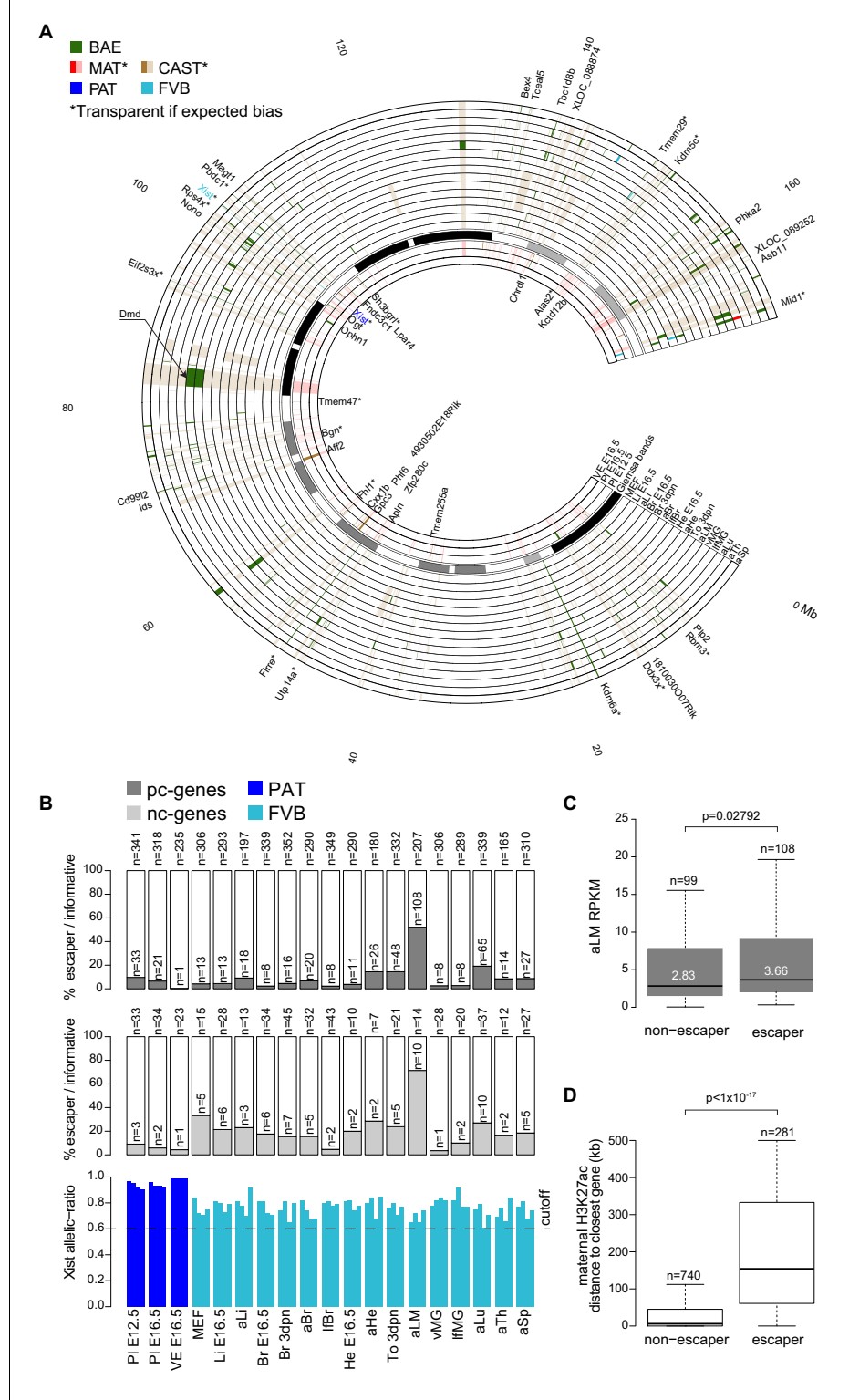

**Figure 3.** X chromosome inactivation (XCI) escapers appear to be highly tissue-specific. (**A**) Circos plot showing the mouse chromosome X Allelome for 19 female tissues (*Figure 1*). Outer layers: 16 embryonic, neonatal and adult tissues showing random XCI (FVB X preferentially inactivated in CAST/FVB F1 tissues (skewed XCI)). Middle layer: Giemsa banding (UCSC genome browser). Inside layers: three extra-embryonic tissues showing imprinted XCI (paternal X chromosome inactivated). The top 25/225 escapers from random XCI are indicated on the outside of the Circos plot, while the top 20/43 escapers from imprinted XCI are indicated on the inside (escapers ranked

*Figure 3 continued on next page*

*Figure 3 continued*

by number of tissues). An asterisk marks known escapers. *Dystrophin* (*Dmd*) escapes specifically in muscle (aLM and To 3dpn, indicated by arrow). Color code as in *Figure 1* except non-escapers are partially transparent (20% opacity, CAST in embryonic and adult tissues, MAT in extra-embryonic tissues). Allelome.PRO settings: FDR 1%, allelic-ratio cutoff 0.6, minread 2. (B) Top: the percentage of pc-genes (black) escaping XCI from all informative pc for each female tissue. The number of informative pc-genes is given above the barplot, while the number of pc escapers is given above each bar. Middle: the same analysis performed for nc-genes (grey). Bottom: the allelic ratio of *Xist* for each replicate in extra-embryonic tissues (*Xist* expressed paternally (blue)) and in non extra-embryonic tissues (*Xist* preferentially expressed from the FVB allele (turquoise)). (C) Leg muscle XCI escapers are expressed at a higher level than non-escapers. Protein coding X-chromosome escaper genes are expressed significantly higher than non-escapers on the X (t-test). Box plots indicate the expression levels of genes in the different categories (outliers not shown). (D) The distance of parental-specific H3K27ac window to the closest non-escaper (331) and escaper (36) gene in placenta E12.5. Maternal H3K27ac windows with a distance higher than 500 kb were not included in the analysis. A boxplot including median values is shown (outliers not shown). After correcting for sample size, a significant difference was observed between escapers and non-escapers (Fisher's exact test, $p<1\times10^{-17}$, details in Materials and methods).

The following figure supplements are available for figure 3:

**Figure supplement 1.** The distribution of X chromosome inactivation escapers across tissues.

**Figure supplement 2.** Validation of X chromosome inactivation escapers in an independent dataset.

---

(*Figure 3A*), but the majority of escaper candidates showed tissue-specific escaping, for example *Dystrophin* (*Dmd*) in muscle (tongue and leg muscle, *Figure 3A*). Ubiquitous and tissue-specific escaping was recently reported using a similar approach to define XCI escapers in the brain, spleen, and ovary (*Berletch et al., 2015*). In our study, we found 123 genes escaping in more than one tissue and 127 escaping in a single tissue, with *Xist* the only gene that escaped in all tissues (*Supplementary file 1*, sheets D-E).

We detected an unexpectedly high number of escapers in leg muscle, with 118 of 221 informative pc- and nc-genes escaping (53.3%), while more than 50 escapers were also detected for tongue and lung (*Figure 3B*). Protein-coding leg muscle escapers showed a significant increase in expression compared to non-escapers (*Figure 3C*). Interestingly, 33 of 53 escapers in the tongue (62%), another muscular tissue, overlapped with escapers found in leg muscle. We found that previously reported XCI escapers from the spleen, brain and placenta that we detected in our data had a higher median SNP coverage than novel XCI escapers, which showed a similar median SNP coverage to leg muscle XCI escapers, further indicating that the sensitivity of our approach may partly explain the high number of escapers detected in leg muscle (*Figure 3—figure supplement 2B*) (*Berletch et al., 2015*; *Finn et al., 2014*). To further test our finding in leg muscle, we re-analyzed adult ₗᵢᵥₑᵣ and leg muscle RNA-seq data from two reciprocal C57BL/6JxCAST F1 crosses from the Gregg lab (*Bonthuis et al., 2015*). We found a similar pattern to our data with 16 of 155 informative genes escaping in liver (10.3%) compared to 86 of 173 informative genes escaping in leg muscle (49.7%), and a high degree of overlap considering the different sequencing protocols and F1 crosses used (*Figure 3—figure supplement 2A*, *Supplementary file 1*, sheet F). Additionally, in E12.5 placenta RNA-seq data that we generated from two reciprocal BALBcx CAST F1 crosses, 15/17 escapers (88%) that we detected were also detected in our previous FVBxCAST analysis, further indicating that our data was reproducible (*Supplementary file 1*, sheet F, the greater number of placenta escapers detected in our FVB/CAST analysis was probably due to the greater sequencing coverage (50 bp single-end versus 100 bp paired-end)).

To investigate if the number of escapers could be explained by different degrees of XCI in different tissues, we examined the *Xist* allelic-ratio (*Figure 3B*, bottom) and expression levels (*Figure 3—figure supplement 1B*, bottom). However, this indicated that all tissues showed the expected imprinted or skewed XCI, and there was no significant correlation between the number of escapers and the *Xist* allelic ratio (Pearson correlation −0.42, p-value=0.068) or *Xist* expression levels (Pearson correlation −0.24, p-value=0.3075). Also correct assignment of all informative genes to the maternal X in XY ES cells (*Figure 1—figure supplement 2D*), and our previous assessment of the STAR

aligner (*Andergassen et al., 2015*), indicated that alignment artifacts do not explain our XCI escaper results.

To determine if differences in allelic H3K27ac enrichment may explain escaper status, we compared the distance to H3K27ac maternal enrichment 4 kb windows in E12.5 placenta, a tissue that shows imprinted XCI of the paternal allele (*Figure 3D*). We found that escapers tended to be further away from the nearest maternal H3K27ac window than non-escapers (p<$10^{-17}$, analysis described in the Materials and methods). This is in agreement with previous reports that *Xist* causes silencing by targeting deposition of repressive chromatin to regulatory elements that then remain marked by H3K27ac on the active allele (*Calabrese et al., 2012*). Given that only *Xist* escapes in all tissues, together our data indicate that tissue-specific escape from XCI may be due to these elements being targeted by *Xist* in a tissue-specific manner.

## Imprinted expression shows tissue-specific regulation

Tissue-specific imprinted expression indicates tissue-specific regulation and that there may be a tissue-specific function for imprinted expression (*Prickett and Oakey, 2012*; *Babak et al., 2015*). Therefore, in order to gain a comprehensive picture of tissue-specific imprinted expression, we used Allelome.PRO to map the mouse Imprintome in our 23 tissues and developmental stages. We previously showed that our total RNA-seq approach combined with Allelome.PRO analysis robustly and sensitively detects imprinted expression in MEFs (*Andergassen et al., 2015*). The Harwell and Otago imprinting databases annotate a total of 126 RefSeq genes, 33 of which are disputed in the literature (downloaded 24th Sept 2015, (*Glaser et al., 2006*; *Williamson et al., 2013*). In our analysis of RNA-seq data, we found 71 of these known genes including five disputed genes (*Pon3, Peg3os, Cd81, Osbpl5,* and *Hymai*) (*Figure 4A*). Three of these genes disputed in placenta (*Pon3, Cd81* and *Osbpl5*) (*Okae et al., 2012*; *Proudhon and Bourc'his, 2010*), we have previously confirmed to show imprinted expression in VE (*Hudson et al., 2011*; *Kulinski et al., 2015*). *Peg3os* and *Hymai* may be false positives due to overlap with known imprinted genes that show the same imprinted expression status, with *Peg3os* overlapped in anti-sense by *Peg3* (due to incomplete strand-specificity of RNA-seq) and *Hymai* overlapped in sense by *Plagl1*. Additionally, although imprinted expression of *Mcts2* could not be assayed by RNA-seq due to a lack of SNPs in our cross, we were able to confirm imprinted expression by differential enrichment of H3K4me3 on its promoter making a total of 70 known imprinted genes when the two probable false positives are removed (*Peg3os* and *Hymai*). The remaining 26 known genes that were not confirmed by RNA-seq fell into five categories: no SNP in the gene body (*Nnat*) or only SNPs in introns (*Xlr3b, Xlr4b, Xlr4c*), an imprinted bias detected below the 0.7 allelic ratio cutoff (*Adam23, Bcl2l1, Zfp64, Casd1, Copg2, Kcnq1, Cobl, Wars, Begain, Dio3, Htr2a*), only detected as biallelically expressed (*Mapt, Ccdc40*), non-informative in all tissues examined (low or no expression, *Htra3, Tfpi2, Zim2, Ins2, Th, 4930524O08Rik,* and *Rhox5*), or the tissue reported to display imprinted expression was not assayed in this study (*Cdh15, Tsix*). In addition to known imprinted genes, this study identified 76 novel candidate imprinted genes (*Figure 4A* and *Figure 4—figure supplement 1*), which required further validation.

We examined our data for the distribution of imprinted expression between tissues and developmental stages (*Figure 4B*). We found the highest number of known imprinted genes in placenta, brain and in neonatal tongue. In general, in extra-embryonic tissues and post-implantation embryonic and neonatal tissues more genes showed imprinted expression than in pluripotent and adult tissues (with the exception of brain). Interestingly, the number of imprinted genes detected tended to decrease within the same tissue during development (see placenta, brain, liver, heart). Tissues important for the energy transfer from the mother to the offspring, the placenta, neonatal tongue and mammary glands, showed a relatively high number of imprinted genes. However, we found a similar pattern in imprinted expression between the brain and mammary glands collected from virgin and lactating females, indicating no obvious role for imprinted expression during lactation.

Tissue-specific imprinted expression could be directly explained by gene expression patterns, with a gene showing imprinted expression wherever it is expressed, or the allelic status of imprinted genes could switch between tissues. To investigate this, we examined the imprinted status across tissues of all known imprinted genes confirmed by RNA-seq in our study (*Figure 4C,D*). This analysis showed that imprinted pc-genes can be categorized into two groups based on the consistency of their allelic status in tissues where they are expressed. The first group ('a') showed variable imprinted expression (in ≤70% of expressing tissues, e.g. *Ago2* and *Slc22a3*), while a second group ('b')

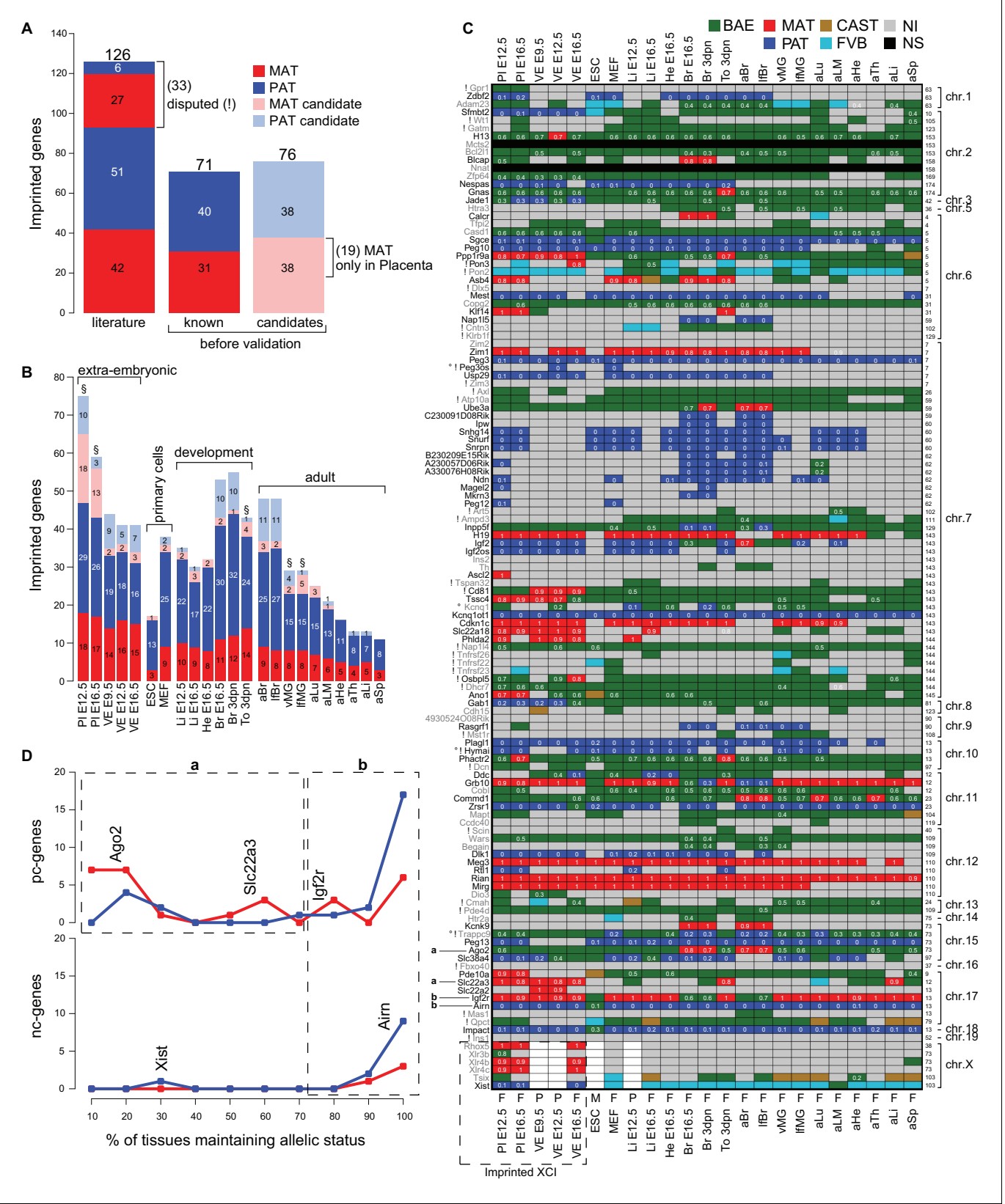

**Figure 4.** The Allelome reveals tissue-specific regulation of imprinted protein-coding genes. (**A**) The number of known and candidate imprinted genes detected in this study by RNA-seq before validation. Known genes were RefSeq genes listed by the Otago or Harwell imprinted databases on 24th

*Figure 4 continued on next page*

*Figure 4 continued*

Sept 2015 (*Glaser et al., 2006*; *Williamson et al., 2013*). (B) The number of known and candidate novel imprinted genes found among different tissues and developmental stages. Tissues important for the energy transfer from the mother to the offspring are indicated (§). (C) The heatmap shows the allelic pattern for all 126 known imprinted genes among the different tissues. Gene names (left side) colored black are confirmed in this study, while gene names colored grey could not be confirmed. ! Indicates disputed genes. ° Probable RNA-seq strand bleed through. Examples given in D for variable ('a', in ≤70% expressing tissues) and consistent ('b' in >70% expressing tissues) imprinted expression are indicated. The chromosome number and the base pair coordinates (in Mb) for each gene are indicated on the right side. The imprinted allelic ratio for each gene and tissue (white) is given only if all four replicates show a bias in the same direction (1 = 100% expression from the maternal allele, 0 = 100% expression from the paternal allele). The sex for each tissue is indicated on the bottom of the heatmap: F (female, XX), M (male, XY), P (pooled XX/XY). Note: allelic analysis of the X chromosome can only be done for female tissues. (D) The percentage of tissues that maintain imprinted expression of protein-coding genes (top) and nc-genes (bottom) (calculated as the number of tissues showing imprinted expression, divided by the number of informative tissues for each gene). The analysis was done for the 69 known imprinted genes confirmed by RNA-seq in this study (*Peg3os* and *Hymai* were excluded due to probable RNA-seq bleed-through). Dotted boxes indicate genes that show variable ('a', in ≤70% expressing tissues) and consistent ('b' in >70% expressing tissues) imprinted expression. Examples are positioned according to the percentage of expressing tissues where they show imprinted expression. Color key as in *Figure 1* except novel maternal and paternal imprinted candidates are pale red and blue, respectively (Allelome.PRO settings: FDR 1%, allelic-ratio cutoff 0.7, minread 2).

The following figure supplement is available for figure 4:

**Figure supplement 1.** The allelic categorization of candidate novel imprinted genes across tissues.

showed consistent imprinted expression (in >70% of expressing tissues, e.g. *Igf2r*). Imprinted nc-genes showed consistent imprinted expression where they were expressed (group 'b', e.g. *Airn*), with the exception of *Xist* whose imprinted expression was restricted to extra-embryonic tissues. In summary, we found that a large proportion of known imprinted pc-genes (47%, group 'a') changed allelic status between tissues indicating tissue-specific regulation of imprinted expression, while in contrast imprinted nc-genes showed imprinted expression wherever they were expressed. This is in agreement with a recent study in human that found that most imprinted genes were tissue-specific and showed biallelic expression in another tissue (*Baran et al., 2015*).

## Novel validated imprinted genes belong to known clusters

Our analysis of RNA-seq data from 23 tissues identified 76 novel imprinted gene candidates that were not present in public databases, and required further validation (*Figure 4A,B* and *Figure 4— figure supplement 1*). We have previously shown that differential enrichment of H3K4me3 over promoters as detected by Allelome.PRO analysis of ChIP-seq data from F1 crosses can validate imprinted expression (*Andergassen et al., 2015*). This has the advantage of being a truly independent technique for validation, since RNA-seq quantifies cDNA as do the commonly used methods for validation of imprinted expression such as quantitative RT-PCR, pyrosequencing, and Sequenom Massarray. Here, we used 4 kb sliding windows for unbiased detection of differential H3K4me3 enrichment for selected tissues to validate novel imprinted genes (*Supplementary file 1*, sheets A, G-J). Using this approach, we were able to validate the X-linked lncRNA *Gm35612*, a MAT expressed imprinted gene in embryonic and adult tissues that we named *CrossFirre*, as it was transcribed anti-sense to *Firre*, a lncRNA involved in regulating nuclear architecture (*Hacisuleyman et al., 2014*) (*Figure 5A*). The previously reported MAT X-linked imprinted genes in brain *Xlr3b*, *Xlr4b*, and *Xlr4c* were classified non-informative in our data due to low SNP coverage (*Raefski and O'Neill, 2005*; *Davies et al., 2005*). *CrossFirre* was only detected as imprinted from RNA-seq data in adult brain where it was relatively highly expressed, while it was non-informative in all other tissues, likely due to difficulty in aligning reads in its repetitive gene body. However, the promoter of *CrossFirre* contains a non-repetitive region, which showed maternal H3K4me3 enrichment in MEFs supporting imprinted expression of *CrossFirre*.

Interestingly, although analysis of RNA-seq data from MEFs and other tissues indicated that *Firre* was biallelically expressed, as expected for a known XCI escaper, in MEFs we found a CAST biased H3K4me3 enrichment on its promoter, in line with the XCI bias in silencing the FVB allele, while the *Firre* gene body contained multiple H3K4m3 peaks enriched for a FVB bias (*Figure 5A*). The existence of multiple alternative TSS within the *Firre* locus was supported by Cufflinks assembly of transcripts matching 6/13 H3K4me3 peaks, and multiple CAGE tags. Interestingly, 10/13 of these

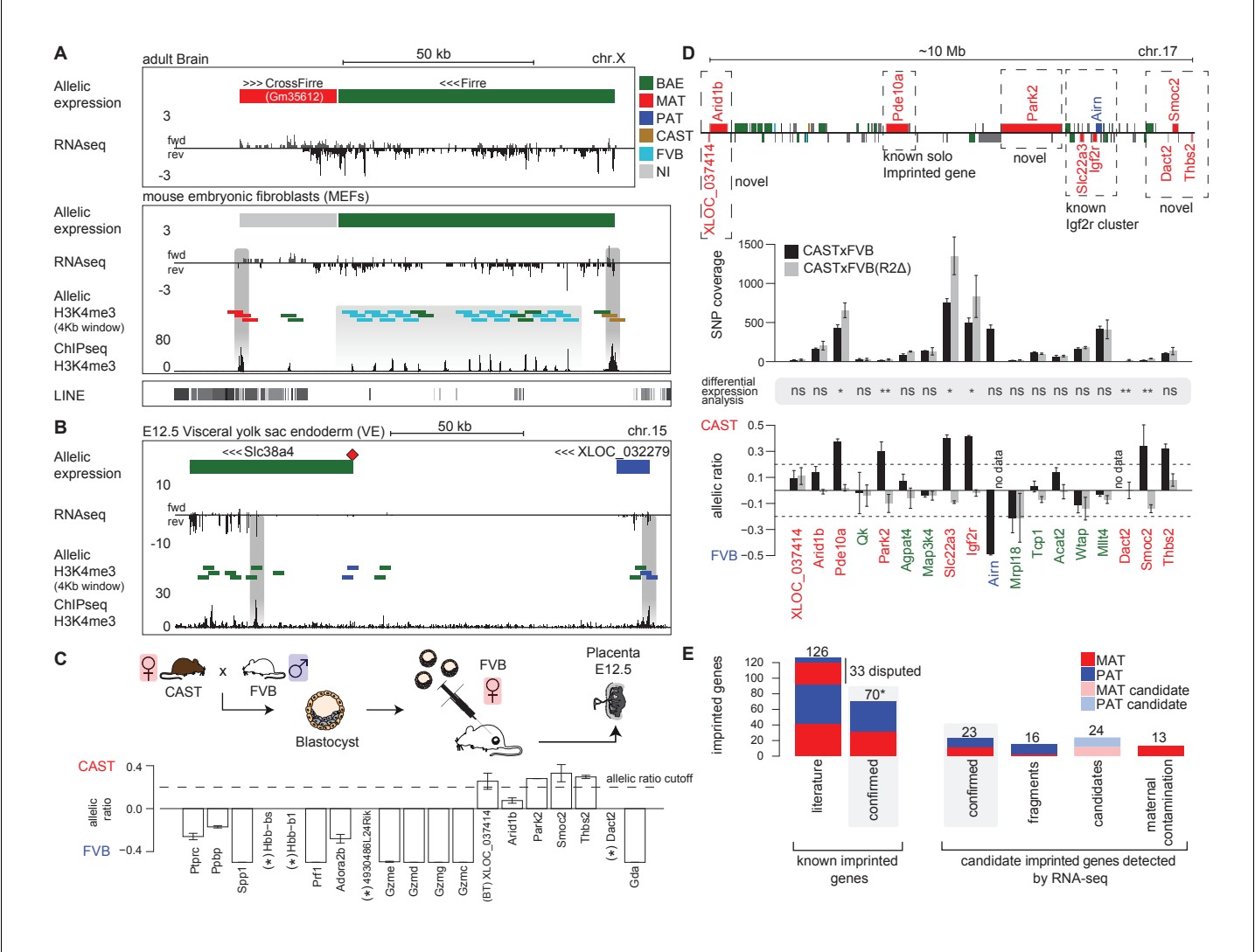

**Figure 5.** Novel validated imprinted genes belong to known clusters. (**A**) A novel X-linked imprinted nc-gene is transcribed anti-sense to *Firre* lncRNA. Maternal imprinted expression of *CrossFirre* (*Gm34612*) detected from RNA-seq in adult brain (top) was validated by maternal H3K4me3 enrichment in MEFs (middle). The gene body of *CrossFirre* is enriched for LINE repetitive elements (bottom). Highlighted in grey are H3K4me3 peaks over the *CrossFirre* promoter (MAT), the canonical *Firre* promoter (CAST), and multiple peaks in the *Firre* gene body (FVB) (UCSC genome browser screenshot). (**B**) *Slc38a4* forms a cluster with a novel imprinted lncRNA. The *Slc38a4* promoter is associated with maternal DNA methylation of the gametic differentially methylated region (gDMR, red square). In E12.5 VE, biallelic expression of *Slc38a4* is associated with biallelic H3K4me3 enrichment over an alternative TSS (highlighted by grey bar). Paternal expression of the novel upstream imprinted lncRNA XLOC_032279 was validated by paternal H3K4me3 enrichment (UCSC genome browser screenshot). (**C**) The allelic ratio in embryo-transferred placentas distinguishes maternal imprinted expression from maternal contamination. CAST (**F**) x FVB (**M**) blastocysts were transferred into pseudo-pregnant FVB females, and placentas collected at E12.5 and subject to ribosome RNA depleted RNA-seq. Genes showing maternal imprinted expression show a bias toward the maternal CAST allele. Genes expressed in maternal blood or decidua (maternal contamination) show a bias toward the FVB allele of the host mother. Novel imprinted genes that showed imprinted expression only in placenta or visceral yolk sac endoderm are displayed. Dotted line indicates 0.7 allelic ratio cutoff, (*) indicates genes with too low SNP coverage to be informative in this analysis. (**D**) The *Airn* promoter deletion (R2Δ) demonstrates that genes over a 10 Mb region are subject to imprinted silencing by *Airn*. First row: Known and novel imprinted genes detected by Allelome.PRO analysis of RNA-seq from E12.5 placenta in 10 Mb region surrounding the known *Igf2r* cluster (UCSC genome browser screenshot). Second row: The SNP coverage (reads over SNPs) of imprinted genes and selected biallelic controls between *Arid1b* and *Thbs2* on chromosome 17 for CASTxFVB and CASTxFVB(R2Δ) E12.5 placentas. Third row: Differential expression analysis calculated using Cuffdiff (version 2.2.1) ** adjusted p-value<0.01* adjusted p-value<0.05, ns non-significant. Fourth row: The allelic ratio (median and standard deviation) for the same genes calculated using the Allelome.PRO pipeline (0.5 = 100% maternal and −0.5 = 100% paternal expression). (**E**) A summary of known and candidate imprinted genes confirmed in this study. After validation, imprinted genes from the literature were confirmed or not (RefSeq genes listed by the Otago or Harwell imprinted databases on 24th Sept 2015 [**Glaser et al., 2006**; **Williamson et al., 2013**]). * 70 confirmed known genes include *Meg3*, *Rian* and *Mirg*, which our data indicates is a single imprinted transcript

*Figure 5 continued on next page*

Figure 5 continued

(*Figure 5—figure supplement 1B*). Novel imprinted candidates were classified as 'confirmed' (validated or supported), 'fragments' (supported nc-genes near known imprinted genes without evidence they are independent transcripts), 'candidates' (detected in one tissue by RNA-seq without further supporting evidence), and 'maternal contamination' (bias toward host mother allele in embryo transfer and/or expression in maternal decidua and blood). The 23 confirmed novel imprinted genes are shown in *Table 1*. Classification of imprinted gene candidates following validation is further explained in the text.

The following figure supplement is available for figure 5:

**Figure supplement 1.** Validation of novel imprinted genes.

internal FVB biased H3K4me3 peaks overlapped with the previously reported RRD repeats, indicating that these repeats may be associated with promoters (*Figure 5—figure supplement 1A*, [*Hacisuleyman et al., 2014*]). Together these data indicate that the *Firre* locus may contain overlapping CAST and FVB biased transcripts but is classified as biallelic when these transcripts are grouped together in the RefSeq annotation.

In addition to confirming the allelic expression status of imprinted genes, H3K4me3 enrichment can indicate the start site of independent transcripts, distinguishing novel imprinted nc-gene candidates from 5′ or 3′extensions of known imprinted genes. In MEFs, a single maternal H3K4me3 peak over the *Meg3* promoter together with continuous transcription and the assembly of transcripts spanning the reported maternally expressed Meg3, Rian and Mirg imprinted lncRNA genes, indicated that Meg3 may form a single long maternally expressed imprinted transcript in this tissue (*Figure 5—figure supplement 1B*). In placenta and VE, we found a novel paternally expressed candidate XLOC_032279 upstream of the known 'solo' imprinted gene in placenta *Slc38a4* that was validated by paternal enrichment of H3K4me3 over its promoter (*Figure 5B*). In E12.5 VE, *Slc38a4* showed biallelic expression due to expression from an alternative downstream promoter, but paternal enrichment over the canonical promoter that is a maternally methylated gametic differentially methylated region (gDMR, red diamond *Figure 5B*) that has not yet been validated as an ICE. This indicated that paternal expression of *Slc38a4* may be masked by a higher level of expression from the biallelic isoform, which was supported by a paternal bias in expression below the allelic ratio cutoff (*Figure 4C*). Furthermore, paternal expression in E9.5 VE and biallelic expression without a bias in E16.5 VE indicated that *Slc38a4* switches from an imprinted to BAE isoform during VE development, while XLOC_032279 maintained imprinted expression at all stages (*Figure 4C* and *Figure 4—figure supplement 1*). These results indicate that XLOC_032279 is an independent imprinted gene that may belong to an imprinted cluster together with *Slc38a4,* which was previously thought to be a solo imprinted gene,.

Maternal imprinted expression in placenta requires validation to distinguish it from expression in maternal decidua, blood and blood vessels that 'contaminate' the placenta. By comparing expression by RNA-seq in decidua with placenta, we found that 11/19 placental-specific maternal imprinted expression candidates had a decidua/placenta ratio >5 indicating they could result from maternal contamination (*Figure 5—figure supplement 1C*). Another three candidates had a low decidua/placenta ratio but are genes expressed specifically in blood (*Ppbp*, *Hbb-bs*, *Hbb-b1*), indicating that they too could result from expression in maternal tissue. To definitively test for maternal contamination, we transferred CAST/FVB F1 blastocysts into FVB host mothers, collected placentas at E12.5 and then performed RNA-seq and Allelome.PRO analysis to determine the allelic expression status of candidate placental-specific maternal imprinted expression candidates (*Figure 5C*). We found that all 19 of the known maternal imprinted genes that we detected in placenta in the initial experiment showed the expected maternal CAST expression bias in the embryo transfer experiment indicating true maternal imprinted expression, although two of these genes, *Ano1* and *Phactr2,* had a maternal bias below the allelic ratio cutoff of 0.7 that we used (*Figure 5—figure supplement 1D*). We found that 10/19 novel maternally expressed candidates had a bias toward the FVB allele of the host mother indicating maternal contamination (*Figure 5C*), all of which were expressed in blood or decidua (*Figure 5—figure supplement 1C*). Interestingly, all five validated novel maternal candidates with a bias toward the maternal CAST allele were in close proximity to the *Igf2r* imprinted cluster (*XLOC_037414, Park2, Smoc2 Thbs2* and *Arid1b (Arid1b* showed a CAST bias below the 0.7

allelic cutoff)), while 4/19 candidates were not informative in this experiment, most likely due to lower sequencing coverage. We further validated these five candidates (all showed a maternal bias although *XLOC_037414* and *Arid1b* were below the 0.7 allelic ratio cutoff), plus two additional maternal imprinted expression candidates (including *Dact2* near the *Igf2r* imprinted cluster) and six paternal imprinted expression candidates by RNA-seq of placentas from a different F1 cross (two reciprocal crosses from BALBc/CAST E12.5 placentas, *Figure 5—figure supplement 1E*).

As noted above, six of the placental maternal imprinted expression candidates, as well as the known solo gene *Pde10a* that shows maternal imprinted expression only in placenta, were in close proximity to the known *Igf2r* cluster on chromosome 17. Therefore, we took advantage of the existing R2Δ mouse model that has a deletion of the *Igf2r* ICE and *Airn* promoter (*Wutz et al., 2001*), to genetically test if these genes are part of the *Igf2r* imprinted cluster (*Figure 5D*). We compared expression in CAST/FVB with CAST/R2D E12.5 placentas and found either a reduction in allelic ratio from maternal biased to biallelic and/or a significant increase in expression for candidates near the *Igf2r* cluster (*Pde10a*, *Park2*, *Dact2*, *Smoc2*, and *Thbs2*, *Arid1b* showed a maternal bias below the 0.7 ratio). *XLOC_037414* had low SNP coverage in this experiment, and did not show a difference in the ratio in the R2Δ mutant. However, the validation of the initial finding of maternal imprinted expression in placenta by differential H3K4me3 enrichment, plus a maternal expression bias in the embryo transfer experiment and BALBc x CAST cross, indicates that *Arid1b* and its bidirectional transcript *XLOC_037414* are true imprinted genes regulated by *Airn*, although the degree of imprinted bias may be less than for other genes in the *Igf2r* cluster. In summary, this experiment demonstrates that *Airn* lncRNA causes imprinted silencing of genes over a 10 Mb region (7.7 Mb upstream and 1.9 Mb downstream of the ICE), extending the size of the *Igf2r* imprinted cluster from its previously known size of 450 kb, and making it the largest *cis*-regulated autosomal region.

Following validation, we were able to confirm 70 known imprinted genes as described above (68 if *Meg3* is a single imprinted transcript overlapping the *Rian* and *Mirg* loci) and 23 novel imprinted genes from the 76 candidates detected by RNA-seq (*Figure 5E*, *Table 1*, *Supplementary file 1*, sheets G-J). These 23 novel imprinted genes had supporting evidence in addition to their initial detection from RNA-seq data, with 16 in the 'validated' category showing either genetic evidence (a loss of imprinted expression following *Airn* promoter deletion indicating a gene belongs to *Igf2r* imprinted cluster), differential H3K4me3 enrichment correlating with imprinted expression (independent technique), embryo transfer supporting maternal imprinted expression (for placenta candidates), and/or detection in BALBc/CAST crosses (different F1 cross). The other seven were in the 'supported' category having been detected in multiple tissues or developmental stages and/or being located within 7 Mb of a gDMR or imprinted region (the gDMR has been shown to be the ICE in all seven tested cases, while 7 Mb was the maximum distance to the ICE that we observed in the expanded *Igf2r* cluster). A total of 5/23 of these novel imprinted genes (*Qk*, *Park2*, *Dact2*, *Fkbp6* and *Platr20*) were recently confirmed by others (*Calabrese et al., 2015*; *Strogantsev et al., 2015*; *Babak et al., 2015*), leaving 18 that were novel to this study. Of the remaining candidates 16/76 were 'fragments', nc-genes with supported imprinted expression, but neighboring known imprinted genes, and with no H3K4me3 peak to support their existence as independent transcripts. Another 24/76 were candidates detected in one tissue by RNA-seq, but without any supporting evidence for their imprinted expression, and 13/76 were false-positive maternal imprinted expression candidates from placenta due to expression in contaminating maternal decidua or blood. Analysis of the genomic location of the 70 known and 23 novel imprinted genes identified here showed that 90.3% are clustered, whereas prior to this study 83.9% of the non-disputed imprinted genes were assigned to clusters (*Glaser et al., 2006*; *Williamson et al., 2013*).

## Tissue and developmental-specific expansion and contraction of imprinted clusters

By combining imprinted expression detected in a comprehensive survey of mouse development we found that 19/23 high confidence novel imprinted genes were located in close proximity to known imprinted genes further indicating that imprinted genes are regulated in clusters (*Figure 6A*). The remaining four novel imprinted genes form new novel imprinted regions. Tissue-specific imprinted expression indicated differences in regulation of imprinted expression, so we compared the size of each imprinted region in each tissue to determine if cluster size changed during development (*Figure 6B*, *Figure 6—figure supplement 1*, *Supplementary file 1*, sheet K). Generally, we found

**Table 1.** Novel imprinted genes. The list of novel imprinted genes detected in this study and confirmed by extra evidence (more detail is given in **Supplementary file 1**, sheet I). Imprinted genes were classified as novel if they were not listed in the Otago or Harwell imprinted gene databases (downloaded 24th Sept 2015, [**Glaser et al., 2006**; **Williamson et al., 2013**]). Previous reports not in these databases are shown.

| Name | Chr | M/P | Validation evidence | Previous reports |
|---|---|---|---|---|
| XLOC_047844 | 2 | PAT | ChIP, BALBc, multiple tissues, proximity | |
| Mafb | 2 | MAT | Proximity | |
| XLOC_050739 | 2 | PAT | ChIP, BALBc, multiple tissues, proximity | |
| 2400006E01Rik | 3 | PAT | BALBc, proximity | |
| Fkbp6 | 5 | PAT | ChIP | (**Strogantsev et al., 2015**) |
| XLOC_075991 | 7 | PAT | sDMR, multiple tissues | |
| XLOC_076143 | 7 | PAT | ChIP, BALBc, multiple tissues, proximity | |
| BC020402 | 10 | PAT | Proximity | |
| XLOC_011039 | 11 | MAT | ChIP, BALBc, multiple tissues, proximity | |
| Platr20 | 11 | PAT | ChIP | (**Babak et al., 2015**) |
| XLOC_011629 | 11 | PAT | ChIP, multiple tissues | |
| Smoc1 | 12 | PAT | ChIP, BALBc | |
| XLOC_032279 | 15 | PAT | ChIP, BALBc, multiple tissues, proximity | |
| Galnt6 | 15 | PAT | Multiple tissues, proximity | |
| XLOC_037414 | 17 | MAT | ChIP, embryo transfer, proximity | |
| Arid1b | 17 | MAT | Genetic, ChIP, proximity | |
| Qk | 17 | MAT | Proximity | (**Calabrese et al., 2015**) |
| Park2 | 17 | MAT | Genetic, embryo transfer, BALBc, multiple tissues, proximity | (**Calabrese et al., 2015**) |
| XLOC_037615 | 17 | MAT | Proximity | |
| Dact2 | 17 | MAT | Genetic, ChIP, BALBc, proximity | (**Babak et al., 2015**; **Calabrese et al., 2015**) |
| Smoc2 | 17 | MAT | Genetic, ChIP, embryo transfer, BALBc, proximity | |
| Thbs2 | 17 | MAT | Genetic, embryo transfer, BALBc, multiple tissues, proximity | |
| CrossFirre (Gm35612) | X | MAT | ChIP | |

Key: Genetic: validation of novel candidates in proximity by genetic deletion of the lncRNA Airn; ChIP: validation by allele-specific H3K4me3 ChIP-seq; sDMR: Transcription start side overlaps sDMRs mapped in Brain (**Xie et al., 2012**); BALBc: validation by RNA-seq in CastxBALBc background; Embryo transfer: Embryo transfer supports maternal expression in placenta; Multiple tissues: Imprinted expression detected in multiple tissues and developmental stages; Proximity: distance between novel candidate and gDMR or known imprinted region < 7 Mb

that cluster size was at a minimum in pluripotent cells, then expanded in post-implantation and extra-embryonic tissues, before retracting to a minimal size in adult tissues (**Figure 6—figure supplement 1**). Exceptions to this were the *Pws/As* and *Kcnk9* clusters where the cluster size in adult brain was equivalent to the maximum that was also found in embryonic and neonatal brain. Interestingly, we observed that 19/28 imprinted regions (68%) showed the maximum size, or equal to the maximum size, in extra-embryonic tissues (**Supplementary file 1**, sheet K). In particular, we noticed that the *Kcnq1* and *Igf2r* imprinted clusters, where imprinted silencing is known to be controlled by an lncRNA, showed a dramatic cluster expansion in extra-embryonic tissues, particularly in placenta (**Figure 6B**).

To investigate how this massive expansion may be regulated, we examined allelic H3K27ac enrichment around imprinted clusters in embryonic liver, and the extra-embryonic VE and placenta (**Figure 6C**). In the *Igf2r* cluster, we found that maternal enrichment of H3K27ac correlated with cluster size, with no enrichment detected for embryonic liver, maternal enrichment windows detected within 2 Mb of the ICE in VE, and up to 7 Mb away in placenta (**Figure 6C**, upper panel). Genome wide we found only a background level of parental-specific H3K27ac enrichment in embryonic liver, whereas VE and placenta showed a significant enrichment of parental-specific H3K27ac in imprinted

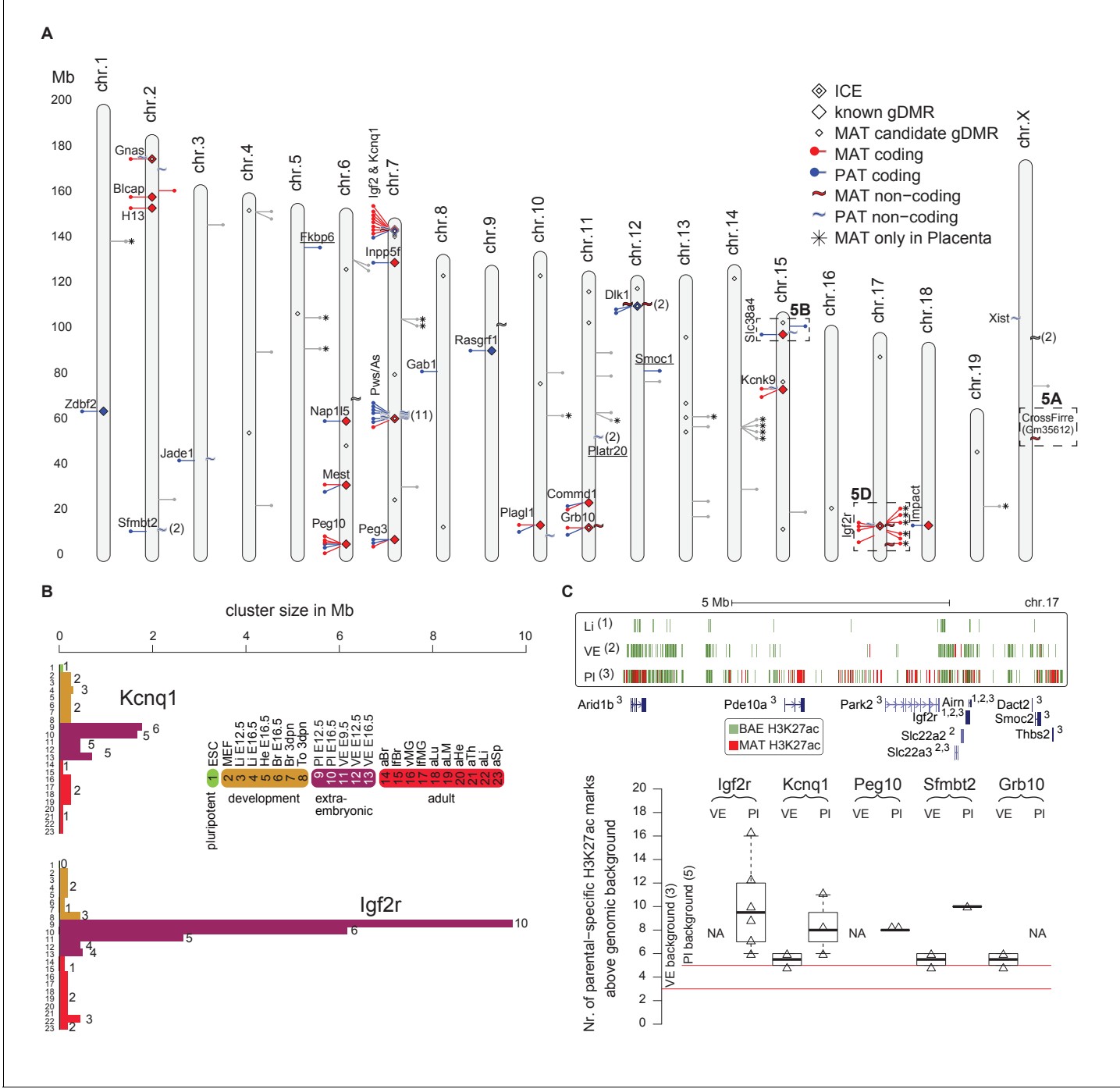

**Figure 6.** Tissue and developmental-specific expansion and contraction of imprinted clusters correlates with parental-specific histone modification. (**A**) A summary of imprinted genes detected in this study. Mouse chromosomes with the positions of known (left side of the chromosome) and novel supported or validated (right side of the chromosome) imprinted pc (—) and nc (~) genes. Candidate imprinted genes that are not supported or validated are indicated in grey. Imprint control elements (ICE), known and candidate gDMRs are indicated (*Proudhon et al., 2012*; *Xie et al., 2012*). * Indicates maternally expressed genes restricted to placenta. The base pair coordinates (Mb) are indicated on the left side. Underlined are new imprinted clusters. Dashed boxes indicate the *Crossfirre* locus shown in *Figure 5A*, the *Slc38a4* cluster shown in *Figure 5B*, and the *Igf2r* cluster shown in *Figure 5D*. Color code as in *Figure 1*. For more details see Materials and methods and *Supplementary file 1*, sheets G-J. (**B**) The *Igf2r* and *Kcnq1* cluster size during development and between tissues (tissue abbreviations as in *Figure 1*). The number of imprinted genes for each developmental stage/tissue is indicated at the top of the bar. (**C**) Top: Allelic H3K27ac enrichment (4 kb sliding windows) over the expanded *Igf2r* cluster for E12.5 Liver, VE and placenta (UCSC genome browser screenshot). Numbers indicate tissue where a gene shows imprinted expression. Bottom: The number of parental-specific H3K27ac 4 kb sliding windows within non-overlapping 100 kb count windows for E12.5 VE and placenta (PI). Counts over the

*Figure 6 continued on next page*

*Figure 6 continued*

background cutoff are shown (defined as the maximum count detected outside of imprinted regions for each tissue). For more details see Materials and methods. NA = not available (no parental-specific windows available for analysis).

The following figure supplement is available for figure 6:

**Figure supplement 1.** Expansion and contraction of imprinted clusters during development.

regions (see Materials and methods for details). Quantifying this for VE and placenta, we found that the extent of H3K27ac expansion correlated with cluster size for these tissues (*Figure 6C*, lower panel).

Altogether, we found that all types of allele-specific expression that we examined were highly tissue-specific. Specifically, we could also distinguish a clear developmental pattern in the numbers of genes showing imprinted expression, with imprinted clusters expanding during development, particularly in extra-embryonic tissues, and then contracting in the adult. For all types of allele-specific expression, we found an association with nearby allele-specific H3K27ac enrichment, indicating that allele-specific expression due to both genetic and epigenetic causes may be mediated through enhancers.

## Discussion

Biases in allelic expression in mammals due to genetic or epigenetic causes can have significant phenotypic consequences, but a comprehensive profile of this has been lacking. Here, using the Allelome.PRO approach that classifies the allelic expression status of all genes in a tissue, we profiled allelic expression in 23 mouse tissues and developmental stages from RNA-seq data. In combination with extensive validation, this provides a valuable resource of strain-biased genes between CAST and FVB, XCI escapers, and together with previous literature, a high confidence list of imprinted genes in mouse, with 93 imprinted genes including 18 genes novel to this study. These data revealed that strain-biased expression, the extent of XCI and imprinted expression were highly tissue-specific. In particular, we show that imprinted gene cluster size varies between tissues and during development, and that they are at their maximum size in extra-embryonic tissues. Interestingly, we found that allelic expression was associated with differential enrichment of H3K27ac in adjacent regions.

Genetic polymorphisms can lead to expression biases in humans, but the outbred nature of the human population makes it difficult to assess the effect the same polymorphism has on allelic expression in different tissues. By using replicates of F1 tissues from crosses of inbred mouse strains, we were able to assess the allelic expression of strain-biased genes in different tissues with the same genetic background. It could be that strain-biased expression is simply a reflection of tissue-specific expression, and that a bias is observed wherever a gene is expressed. However, we found that more often strain-biased genes showed a switch in allelic status between tissues, that in a sub-set of cases correlated with an apparent switch in enhancer usage with different allelic biases, indicating that strain-biased expression may result from genetic differences in tissue-specific enhancers that control tissue-specific expression (*Leung et al., 2015*). Our results indicated that other strain-biased expression may be explained by allele-specific differences in post-transcriptional stability, which could lead to tissue-specific strain-biased expression due to tissue-specific expression of miRNAs or RNA binding proteins that bind and affect the stability of the transcript (*Gaidatzis et al., 2015*; *Geuvadis Consortium et al., 2013*).

*Xist* expression leading to XCI and the parental-allele-specific DNA-methylated ICEs that control imprinted expression are present in almost every cell type during development, so it might be expected that if a gene subject to epigenetic silencing by these processes is expressed then it would always be silenced on one allele. However, this is not the case. We found that so-called ubiquitous XCI escapers that escape in many tissues were in the minority, with most XCI escapers escaping silencing only in 1 or 2 tissues. Tissue-specific and ubiquitous XCI escapers were previously observed (*Berletch et al., 2015*), but our study encompassing 19 females tissues enabled us to obtain a more accurate picture of the tissue-specific nature of XCI escapers. Similarly, a high proportion of

imprinted genes showed tissue-specific imprinted expression where they switched to BAE in another tissue. These results showed that biases in allelic expression are generally tissue-specific, whether they arise from genetic or epigenetic causes, indicating that tissue-specific features are responsible for switches in allelic status. Interestingly, we found a novel maternally expressed gene on the X chromosome called *CrossFirre*, that as the only X-linked imprinted gene that we validated, may warrant further investigation for a connection to imprinted XCI.

Related to the variable tissue-specific imprinted expression, we found that the size of imprinted clusters varied during development, showing a minimum size in pluripotent ESC and a maximum in extra-embryonic tissues. These results have interesting parallels with XCI, with ESC showing two active alleles prior to the onset of random XCI, while extra-embryonic tissues are the only post-implantation tissues to show imprinted XCI (*Wutz, 2011*). Specifically, we showed that the *Igf2r* imprinted cluster is much larger than previously thought extending over 10 Mb in placenta, or around 10% of mouse chromosome 17. The scale of the region affected by imprinted silenced by *Airn* is reminiscent of XCI by *Xist*. Early in XCI *Xist* recruits PRC1 and PRC2 (*Wutz, 2011*), repressive histone modifying complexes that have also been associated with *Airn* (*Terranova et al., 2008*), further indicating that they may act by a similar mechanism to cause silencing of distant genes.

The H3K27ac histone modification marks open chromatin and has been associated with active enhancers (*Creyghton et al., 2010*). Following this we found that a switch from strain-biased to BAE between tissues may be explained by tissue-specific enhancer usage associated with the corresponding H3K27ac enrichment. We also found an association between allelic H3K27ac enrichment and genes subject to XCI and imprinted silencing. *Xist* is reported to target H3K27me3 deposition to regions that remain marked by H3K27ac on the active allele (*Calabrese et al., 2012*). Consistent with this, we found that the distance to allele-specific enrichment of H3K27ac was greater for XCI escapers than for genes subject to XCI. Enhancers explain tissue-specific expression, so it follows that tissue-specific silencing seen for XCI and imprinted silencing may be explained by actions on tissue-specific enhancers. Following this we found that the size of an imprinted cluster in a particular tissue correlated with size of the region showing parental-allele specific H3K27ac enrichment. Together these results indicate that all types of allele-specific expression that we observed may be mediated by allele-specific actions on enhancers.

## Materials and methods

### Mouse strains

Mice were bred and housed according to Austrian regulations under Laboratory Animal Facility Permit MA58-0375/2007/4. FVB/NJ (FVB) and BALBc mice were purchased from Charles River and CAST/EiJ (CAST) from the Jackson Laboratory. Embryo transfer was performed according to standard procedures, and was approved by the IMP/IMBA animal ethics committee (*Behringer et al., 2014*). The FVB.129P2-Airn<R2D> (R2Δ) mouse has a deletion that includes the *Airn* promoter and the imprint control element (ICE) of the *Igf2r* imprinted cluster (*Wutz et al., 2001*). F1 tissues were collected in replicates and frozen and stored at −80°C until further processed. Further details are provided in *Supplementary file 1*, sheet A.

### Cell lines

Cell lines used in this study have been previously characterized and published elsewhere, and were all mycoplasma free. Primary mouse embryonic fibroblasts (MEFs) were derived from reciprocal FVB x CAST crosses from E12.5 embryos (*Andergassen et al., 2015*). Mouse ES cells were derived from reciprocal crosses from FVB x CAST crosses from mouse blastocysts (*Kulinski et al., 2015*). The passage number is given for each cell line used, and for each experiment in *Supplementary file 1*, sheet A.

### Tissue isolation for RNA-seq

To determine the mouse expression Allelome from RNA-seq data, we collected samples from 23 F1 mouse tissues and developmental stages (2x CASTxFVB and 2x FVBxCAST, maternal allele always on the left) representing pluripotent (1), embryonic (5), extra-embryonic (5), neonatal (2), adult (8) and lactating female (2) tissues (*Figure 1A*, *Supplementary file 1*, sheet A). We collected 19 tissues

samples from individual females (XX), while for embryonic day (E) 12.5 liver, and E9.5 and E12.5 visceral yolk sac endoderm (VE) we pooled males and females from one litter (XX/XY). Embryonic stem cells (ESCs) were derived from male (XY) clones. Sex was confirmed by PCR for individual extra-embryonic, embryonic and neonatal tissues samples (*Capel et al., 1999*).

Tissues were dissected and frozen immediately in liquid nitrogen, with the exception of ESCs, MEFs, VE and mammary glands that were processed differently before freezing and storing at −80°C. For ESCs and MEFs, cells were centrifuged, washed in PBS and then frozen. Visceral yolk sac and mammary glands were processed as previously described to isolate VE and mammary epithelial cells (*Hudson et al., 2011*; *Joshi et al., 2010*). ESCs were derived following an established protocol (*Bryja et al., 2006*; *Kulinski et al., 2015*), and adapted to 2i media without feeders (ESGRO-2i Medium, Millipore) (*Ying and Smith, 2003*; *Ying et al., 2008*). Note that the RNA-seq data from MEFs was described in an earlier study (*Andergassen et al., 2015*).

To determine genes subject to imprinted silencing by *Airn* long non-coding (lnc) RNA and exclude maternal contamination, we crossed CASTxR2Δ and collected placentas from three wild type (WT) and three mutant embryos from embryonic day (E) 12.5, as well as 3x CAST decidua from WT embryos.

## Tissue isolation for ChIP-seq

For ChIP-seq experiments, we collected material from FVBxCAST reciprocal crosses. Individuals were collected for adult liver, placenta, mammary glands and neonatal tongue, while samples were pooled for embryonic liver and VE (for details see *Supplementary file 1*, sheet A). Note that the ChIP-seq data from MEFs was described in an earlier study (*Andergassen et al., 2015*).

## RNA and ChIP-seq

RNA was extracted from TRI-reagent using standard protocols (Sigma-Aldrich T9424). DNaseI treated (DNA-Free Ambion) total RNA (1–3 µg) was depleted for Ribosomal RNA using the RiboZero rRNA removal kit (Human/Mouse/Rat, Epicentre) or enriched for polyA containing mRNA (Illumina). Strand-specific RNA-seq libraries were generated using the TruSeq RNA Sample Prep Kit v2 (Illumina) modified as previously described (*Sultan et al., 2012*). Native ChIP for H3K4me3 and H3K27ac was performed as previously described (*Regha et al., 2007*). The TruSeq ChIP Sample Prep Kit (Illumina) was then used to prepare ChIP-seq libraries. RNA-seq and ChIP-seq was then performed on a Illumina HiSeq. For further details see *Supplementary file 1*, sheet A.

## Preparation of the input files

The RNA and ChIP-seq data was aligned with STAR (*Dobin et al., 2013*) as previously described (*Andergassen et al., 2015*). For our standard analysis, we only used uniquely aligned reads. In order to make a fair comparison between tissues, we equalized the number of uniquely aligned reads used for Allelome.PRO analysis of total RNA-seq from the 23 tissues, and H3K27ac ChIP-seq from E12.5 embryonic liver, VE and placenta. For each tissue, we took the number of reads from the start of the unaligned FASTQ file required to obtain approximately 15 million uniquely aligned pairs (30 million reads) per sample for RNA-seq data, and 20 million uniquely aligned single end reads for H3K27ac ChIP-seq data. We then realigned these reads allowing only uniquely aligned reads, and performed all subsequent analysis on this data (*Supplementary file 1*, sheet B). For the analysis of the CAST/FVB placenta, CAST/R2Δ placenta and CAST decidua polyA RNA-seq samples we used approximately 18 million uniquely aligned single end reads per sample. For H3K4me3 ChIP-seq data all available reads were analyzed.

## RNA-seq annotation file

To define allelic expression using the Allelome.PRO pipeline, we downloaded the RefSeq annotation from the UCSC genome browser (GRCm38/mm10) on July 15th 2015 and removed transcripts with a gene body length less than 100 bp. To annotate the regions not covered by RefSeq, we combined the reads from the four samples for each tissue using SAMtools (version 1.2) and used Cufflinks (version 2.2.1) to perform a reference based assembly. Next we used Cuffmerge to merge the assemblies from all tissues together with the RefSeq annotation (-g RefSeq.gtf). Then we discarded transcripts overlapping RefSeq in sense orientation and single exon transcripts.

To predict whether the novel annotated transcripts are protein-coding or non-coding, we used the Coding Potential Calculator (CPC) based on sequence features (*Kong et al., 2007*), modified as described previously (*Kornienko et al., 2016*), and RNA-code based on evolutionary signature (*Washietl et al., 2011*). We used the two pipelines for each transcript in the annotation (n = 171389) and assigned the smallest CPC and RNA-code score to each locus. A t-test was performed between mRNAs (RefSeq NM), non-coding RNAs (RefSeq NR) and genes annotated in this study (XLOC) for the CPC and RNA-code score. As expected, we observed a highly significant difference between RefSeq mRNAs (NM) and RefSeq non-coding RNAs (NR) for both the CPC and the RNA-code score (CPC t = 56.4326 [2.239631;2.400904], p-value<$2.2\times10^{-16}$, RNAcode t = 39.6171 [34.44711;38.03595], p-value<$2.2\times10^{16}$). The same significant difference was observed between mRNAs (NM) and genes annotated in this study (XLOC) (CPC t = 72.1867 [2.22801;2.35240], p-value<$2.2\times10^{16}$, RNAcode t = 50.8467 [33.86692;36.58337], p-value<$2.2\times10^{16}$. However, we observed no significant difference between RefSeq non-coding RNAs (NR) and genes annotated in this study (XLOC) showing that the bulk of novel annotated genes is non-coding. (CPC p=0.5079, RNAcode p=0.3504, *Figure 1—figure supplement 3*).

The final annotation consists of 23,521 RefSeq genes (20743 protein-coding (NM) and 2778 NR non-coding) and 6290 assembled non-coding genes (XLOC) outside of the RefSeq annotation.

## ChIP-seq annotation files

Sliding windows were used to define allelic ChIP (4 kb sliding windows for H3K27ac and H3K4me3 (2 kb intervals)).

## SNP annotation files

The SNP annotation file containing 20,601,830 high confidence SNPs between the CAST/EiJ and FVB/NJ strains was extracted from the Sanger database as described previously (*Andergassen et al., 2015*; *Keane et al., 2011*). For RNA-seq, but not ChIP-seq, SNPs overlapping retroposed genes including pseudogenes (RetroGenes V6 from UCSC genome browser) were removed leaving 20,453,039 SNPs. For the CAST x FVB.129P2-Airn-R2D (R2Δ) cross we used only CAST/FVB SNPs where the FVB allele was shared with all three sequenced 129 strains (16,988,479 SNPs).

## Allelome.PRO analysis of RNA and ChIP-seq data

Allele-specific expression and histone modification enrichment was detected from RNA-seq and ChIP-seq data using the Allelome.PRO program as previously described in detail (*Andergassen et al., 2015*). Briefly, for each tissue, a gene or region was classified as showing an imprinted or strain bias if all replicates showed the same direction of bias, the allelic score of all biological replicates passed the FDR cutoff (allelic score was defined as the $\log_{10}(p)$ value calculated from deviations from the binomial distribution of summed reads, the FDR was defined by mock comparisons of the allelic score), and the median allelic ratio was above or equal to the allelic ratio cutoff. Informative genes that did not fulfill these criteria were classified as biallelic. A gene was classified as informative in a given tissue if a minimum SNP coverage was reached for all replicates. This was defined as the minimum SNP coverage required to pass the FDR allelic score cutoff assuming that the allelic ratio would be equal to the allelic ratio cutoff.

For each annotated region, two allelic scores were calculated, a strain biased score (s.score, calculated by comparing summed FVB and CAST reads) and a parental biased or imprinted score (i.score, calculated by comparing summed maternal and paternal reads), and assigned a positive or negative value based on the direction of bias (MAT >0, PAT <0, imprinted score, CAST >0, FVB <0 strain biased score). A summary i.score and s.score were calculated for each gene or region by taking the minimum score of the replicates when each replicate showed the same direction of bias, and setting the summary score to 0 when any replicates showed disagreement. Therefore, each gene or region had either a summary i.score or summary s.score with the other score set to 0, or both summary allelic scores were set to 0.

For allelic analysis with an allelic ratio cutoff of 0.7 (all analysis except the X chromosome inactivation (XCI) escaper analysis), the minimum SNP coverage required for a gene to be called informative varied between tissues from 11 to 13 reads, with a mean of 12 reads. For the XCI escaper analysis

with an allelic ratio cutoff of 0.6, the minimum SNP coverage required for a gene to be called informative varied between tissues from 13 to 48 reads, with a mean of 27 reads.

For RNA-seq informative genes had a mean of 49 informative SNPs with a minread parameter of 2.

The Allelome.PRO settings used in this study:

Allelome RNA-seq: FDR 1%, allelic ratio cutoff 0.7, minread 2

XCI escaper RNA-seq: FDR 1%, allelic ratio cutoff 0.6, minread 2

CASTxR2Δ RNA-seq: FDR 1%, minread 1 (no allelic ratio cutoff)

Imprinted gene validation, H3K4me3 ChIP-seq: FDR 1%, allelic ratio cutoff 0.7, minread 2

H3K27ac ChIP-seq enrichment: FDR 1%, allelic ratio cutoff 0.7, minread 1

(Note: minread = minimum number of reads that must cover a SNP for it to be included in the analysis).

## Calculating enrichment of H3K27ac near strain-biased genes that switch allelic status

Informative H3K27ac 4 kb windows were extracted from the Allelome.PRO output for E12.5 liver and VE. Windows mapping to the X chromosome, and windows overlapping ±2 kb of the transcription start side (TSS) of all RefSeq isoforms and non-coding loci were removed using BEDtools (version 2.20.1)(*Quinlan and Hall, 2010*). The remaining H3K27ac windows were assigned to genes in our annotation if they were within ±50 kb of the TSS (genes without SNPs were excluded, a window could be associated with more than one gene). For informative windows assigned to genes, we calculated the distance to the TSS (upstream (-) or downstream (+) taken from the middle of the window). We then shuffled the allelic status of the H3K27ac windows 100x to generate a random dataset that we subsequently used to calculate enrichment over random.

We selected a subset of genes for further analysis that showed strain-biased expression for CAST or FVB in liver or VE that then switched to BAE in the other tissue. In addition, we called H3K4me3 peaks using MACS and performed an inner join (multiIntersectBed) from the four replicates for each tissue (*Zhang et al., 2008*). Next we removed strain-biased switchers where H3K4me3 peaks did not overlap the promoter (±2 kb of the annotated TSS) in both tissues. For these CAST (53 pc and two nc-genes) and FVB (82 pc and two nc-genes) switchers, we then calculated H3K27ac allelic enrichment over random for each category (BAE, CAST, FVB) ±50 kb from the TSS, when showing strain-biased or biallelic expression. The number of windows detected for each category were counted in 4 kb bins over ±50 kb from the TSS, and enrichment over random calculated for each bin by dividing this number by the mean count for this category from the 100x shuffled allelic tags for the same genes. The H3K27ac enrichment was then plotted for BAE, CAST and FVB for each expression status (CAST, BAE (switching from CAST), FVB and BAE (switching from FVB) (*Figure 2C*). To test for significant enrichment we performed a t-test comparing the 25 bins from the real data to the mean of the random data from the 25 bins (*Supplementary file 1*, sheet C).

## Distinguishing transcriptional and post-transcriptional causes of strain-biased expression by assessing intronic and exonic allelic biases

First, we took strain-biased RefSeq genes detected in E12.5 MEF, fetal liver and VE where we had H3K4me3 data, and kept genes with H3K4me3 enrichment over their annotated TSS that was either BAE or monoallelic matching their strain biased expression. Second, we downloaded the RefSeq gene annotation from introns and exons separately using the USCS table browser, and extracted exonic and intronic SNPs using BEDtools (intersect). Allelome:PRO was then used to calculate allele-specific expression for intronic and exonic reads separately using these SNP files and the same gene annotation as for other analysis. We then kept only genes that were classified as strain-biased or BAE in their exons and introns. Finally, we merged data from strain-biased genes with BAE or monoallelic H3K4me3 enrichment on their promoter (step 1), with strain-biased genes that were classified as BAE or monoallelic in both their exons and introns (step 2), and kept the overlap. We then plotted the allelic ratio of introns versus exons separately for those strain biased genes with BAE H3K4me3 on their promoter, and those with monoallelic H3K4me3 enrichment on their promoter (scatterplot R, *Figure 2E*).

## Detecting X chromosome inactivation escapers

We detected X chromosome escapers as genes that deviated from the expected maternal bias in imprinted X chromosome inactivation (XCI) in extra-embryonic tissues, or the expected CAST bias due to skewed XCI in CASTxFVB crosses in other tissues. To increase the stringency in defining XCI escapers, we used a lower allelic ratio cutoff of 0.6 for the Allelome.PRO analysis, compared to all other analysis in this study (note: genes below the allelic ratio cutoff are classified biallelic, and therefore escapers). In addition, we excluded genes where all four replicates showed the expected MAT or CAST bias and a median allelic ratio above the cutoff, but were classified biallelic by Allelome.PRO because one or more replicates were below the FDR cutoff due to low expression. This approach enabled us to avoid setting an arbitrary RPKM cutoff for escapers.

## Validation of the adult leg muscle XCI escapers

Next, we tested if XCI escapers in leg muscle displayed the expected doubling of expression compared to the non-escapers (*Figure 3C*). We observed a significant increase (t = −2.2184 [−12.0936286; −0.7026719], p-value=0.02792, Cohen's d = 0.3007881) in the expression level of protein-coding escapers compared to non-escapers.

## Distance to H3K27ac maternal windows for XCI escapers and non-escapers in placenta

Maternal H3K27ac 4 kb windows mapping to the X chromosome were extracted from the Allelome.PRO output for placenta E12.5. Windows overlapping ±2 kb of the TSS of all RefSeq isoforms and non-coding loci were removed using BEDtools (version 2.20.1)(*Quinlan and Hall, 2010*). For each maternal window (744 windows), the distance to the nearest escaper (36) and non-escaper (331) genes in E12.5 placenta was calculated using the Bedtools parameter closest 'first'. Maternal H3K27ac windows with a distance higher than 500 kb were excluded from the analysis. Distances to the nearest maternal H3K27ac window were then plotted as a boxplot for both escapers and non-escapers (*Figure 3D*).

To determine if the greater distance to the nearest maternal H3K27ac window observed for escapers was significant, we applied a statistical approach to correct for sample size. We compared the distance to the nearest maternal H3K27ac window for the 36 escapers with 36 non-escapers chosen randomly from the 331 non-escaper genes, and calculated a p-value (t-test). This was repeated a total of 10x, and then the p-values were combined using Fishers's exact test (sumlog method from the metap package in R). This indicated that H3K27ac maternal windows were significantly more distant from escapers than non-escapers (Fisher's exact test, $p<1\times10^{-17}$, p-values corrected for multiple testing using p.adjust function in R (method: fdr))

Reference: Michael Dewey (2014). metap: Meta-analysis of significance values. R package version 0.6. http://CRAN.R-project.org/package=metap

## Published list of known imprinted genes

The list of 126 known imprinted genes was constructed by merging the Harwell and Otago imprinted databases and removing genes not annotated in RefSeq (http://www.mousebook.org/imprinting-gene-list, www.otago.ac.nz/IGC (downloaded 24th Sept 2015, *Glaser et al., 2006*; *Williamson et al., 2013*).

## Detection of alternative transcripts at the *Firre* locus

To detect transcripts at the *Firre* locus we re-aligned the MEF RNA-seq data using the standard settings of STAR (20 multi-mappers allowed compared to our standard analysis where we only allowed uniquely aligned reads) and assembled transcripts with Cufflinks using sensitive settings (cufflinks -j0 -F0 -p10 –library-type fr-firststrand),

## Detection of genes subject to imprinted silencing by *Airn*

To determine if novel imprinted genes detected in placenta near the *Igf2r* cluster belonged to the cluster, we examined whether imprinted silencing of these genes was regulated by *Airn*. Paternal deletion of the imprint control element (ICE) and *Airn* promoter (R2Δ) results in loss of imprinted expression for all genes in the *Igf2r* cluster (*Wutz et al., 2001*). Therefore, we compared the

expression and allelic ratio calculated by Allelome.PRO of the novel candidates between RNA-seq for 3x CAST/FVB and 3x CAST/R2Δ E12.5 placentas (*Figure 5C*, *Supplementary file 1*, sheet A). Differential gene expression was calculated using Cuffdiff (version 2.2.1) to compare the CASTxFVB and CASTxR2Δ samples and the q-value (corrected p-value) plotted (* $0.05 \geq$ q-value $>0.01$, ** $0.01 \geq$ q-value $>0.001$,*** $0.001 \geq$ q-value). The allelic ratio for each replicate and genotype (CASTxFVB and CASTxR2Δ) was calculated using the Allelome.PRO pipeline, and then the mean and standard deviation was plotted. For the Allelome.PRO analysis, we used a SNP annotation filtered for CAST/FVB SNPs where the FVB allele was shared with all three sequenced 129 strains (16,988,479 SNPs). This was necessary as the R2Δ allele was made on a 129 background, and therefore the region near the *Igf2r* may still be of 129 origin.

## Validation of novel imprinted candidates

Following validation we classified the novel imprinted candidates into four categories (*Supplementary file 1*, sheets G-J):

I. Candidate: Detected in one tissue by RNA-seq (Note: maternal placenta candidates were excluded from this category as they required further evidence)

II. Supported Candidate: Detected in multiple tissues or developmental stages (placenta candidates in different tissues), detected in BALBc cross, or located near a known imprinted region (<7 Mb, distance defined in this study based on the distance of *Arid1b* to the ICE in the *Igf2r* cluster).

III. Validated Candidate: Imprinted expression confirmed by parental-specific H3K4me3 ChIP-seq enrichment on the promoter (*Supplementary file 1*, sheet G), maternal imprinted expression confirmed in embryo transfer experiment, or by showing a loss of imprinted expression by deleting the ICE as demonstrated for the *Igf2r* cluster.

IV. Maternal Contamination (candidate excluded): Genes showing maternal expression restricted to placenta with a Decidua/Placenta expression ratio >5, and not supported by any other validation method were defined as maternal contamination. Additionally, blood-specific genes detected as maternally expressed in placenta were also classified as maternal contamination and excluded.

Note: lncRNA candidates were defined as fragments if they were in proximity to a known imprinted lncRNA that showed the same allelic status, and were not supported as an independent transcript by H3K4me3 enrichment on their promoter. Fragments were classified into the same four validation categories as independent transcripts.

## Parental-specific H3K27ac enrichment within imprinted regions

Autosomal parental-specific H3K27ac 4 kb sliding windows were extracted from the Allelome.PRO output for placenta and VE. Windows overlapping ±2 kb of the TSS of all RefSeq isoforms and non-coding loci were removed using BEDtools (version 2.20.1) (*Quinlan and Hall, 2010*). Next, we counted the overlap of parental-specific H3K27ac windows with 100 kb non-overlapping count windows using the count function of BEDtools (intersect –c). For each tissue, we generated a BED file of the imprinted regions based on the most distal and proximal known or novel imprinted genes or fragments (validated or supported) detected for each imprinted region in that tissue (*Supplementary file 1*, sheet K). For each tissue, the imprinted region BED file was joined with the associated BED file containing the parental-specific window counts using BEDtools (intersect). Next, we compared 100 kb count windows that contained at least one 4 kb parental-specific window count. In VE, 14 informative 100 kb windows (mean of 3.357 4 kb windows) were overlapping imprinted regions while 13 were outside (mean = 1.230 4 kb windows). This indicates parental-specific H3K27ac is significantly enriched in imprinted regions (t = 6.5903, [1.605559; 3.061107], p-value=$5.466139^{-07}$, Cohen's d = 1.2683). In placenta, 65 informative 100 kb windows (mean = 3.43 4 kb windows) overlapped imprinted regions while 3737 were outside (mean = 1.195 4 kb windows), indicating a significant enrichment of H3K27ac in imprinted regions in placenta (t = 109.669, [1.211767; 1.255882], p-value<$2.2^{-16}$, Cohen's d = 1.778607).

The maximum count outside of imprinted regions was then used as cutoff to define the background (VE = 3 and Pl = 5). The remaining 100 kb count windows within imprinted regions were then plotted in R (*Figure 6C*).

## Data access

All sequencing data were deposited at the NCBI GEO data repository under accession numbers GSE75957 and GSE69168.

The analyzed data can be viewed on the UCSC genome browser and scripts used in the analysis can be downloaded at the following link: https://opendata.cemm.at/barlowlab/.

## Acknowledgements

This work was partly supported by the Austrian Science Fund (FWF P25185-B22, FWF F4302-B09, FWF W1207-B09). High-throughput sequencing was conducted by the Biomedical Sequencing Facility (BSF) at CeMM in Vienna. Rita Casari and Philipp Guenzl helped with tissue collection. We thank Anton Wutz for critical reading of an earlier version of the manuscript. Johannes Tkadletz from the IMP/IMBA Graphics Service and Pieter-Jan Volders helped with figure production.

## Additional information

### Funding

| Funder | Grant reference number | Author |
|---|---|---|
| Austrian Science Fund | P25185-B22 | Quanah J Hudson |
| Austrian Science Fund | F4302-B09 | Denise P Barlow |
| Austrian Science Fund | W1207-B09 | Denise P Barlow |

The funders had no role in study design, data collection and interpretation, or the decision to submit the work for publication.

### Author contributions

DA, Conceptualization, Software, Formal analysis, Investigation, Visualization, Methodology, Writing—original draft, Writing—review and editing; CPD, Software, Formal analysis, Investigation, Visualization, Methodology, Writing—review and editing; DW, VS, PCB, H-CT, Investigation, Methodology; MM, DM, TMK, Investigation; JMP, Supervision; CB, Supervision, Writing—review and editing; DPB, Conceptualization, Supervision, Funding acquisition, Visualization, Writing—original draft, Writing—review and editing; FMP, Conceptualization, Software, Formal analysis, Supervision, Investigation, Visualization, Writing—original draft, Writing—review and editing; QJH, Conceptualization, Formal analysis, Supervision, Funding acquisition, Investigation, Visualization, Writing—original draft, Writing—review and editing

### Author ORCIDs

Daniel Andergassen, http://orcid.org/0000-0003-1196-4289
Josef M Penninger, http://orcid.org/0000-0002-8194-3777
Christoph Bock, http://orcid.org/0000-0001-6091-3088
Quanah J Hudson, http://orcid.org/0000-0002-3407-4388

### Ethics

Animal experimentation: Mice were bred and housed at the IMBA/IMP facility in Vienna in strict accordance with national recommendations under Laboratory Animal Facility Permit MA58-0375/2007/4

## Additional files

### Supplementary files

• Supplementary file 1. Validation of X chromosome inactivation (XCI) escapers and imprinted expression. (A) Samples analyzed in this study. (B) Equalizing the input reads between samples. Input reads were adjusted to obtain approximately 15Mio uniquely aligned fragments for RNA-seq (30Mio single reads) and approximately 20Mio uniquely aligned reads for H3K27ac ChIP-seq per replicate.

(C) Significant enrichment of allelic H3K27ac based on 100x random permutations of the allelic tags (D) X chromosome inactivation (XCI) escaper candidates from epiblast-derived tissues (embryonic and adult tissues that show skewed XCI in CAST/FVB). (E) XCI escaper candidates from extra-embryonic tissues that show imprinted XCI. (F) XCI escaper validation. (G) Validation of imprinted expression detected by RNA-seq using H3K4me3 ChIP-seq differential enrichment on the promoter (4 kb window). (H) Validation of imprinted expression. (I) Confirmed and novel validated independent imprinted genes. (J) Summary of novel imprinted gene candidates. (K) Tissue and developmental specific expansion and contraction of imprinted clusters.

### Major datasets

The following dataset was generated:

| Author(s) | Year | Dataset title | Dataset URL | Database, license, and accessibility information |
|---|---|---|---|---|
| Andergassen D, Pauler FM, Hudson QJ | 2017 | Mapping the mouse Allelome reveals tissue-specific regulation of allelic expression | http://www.ncbi.nlm.nih.gov/geo/query/acc.cgi?acc=GSE75957 | Publicly available at the NCBI Gene Expression Omnibus (accession no: GSE75957) |

The following previously published dataset was used:

| Author(s) | Year | Dataset title | Dataset URL | Database, license, and accessibility information |
|---|---|---|---|---|
| Andergassen D, Dotter CP, Hudson QJ, Pauler FM | 2015 | Allelome.PRO A pipeline to define allele-specific genome features | https://www.ncbi.nlm.nih.gov/geo/query/acc.cgi?acc=GSE69168 | Publicly available at the NCBI Gene Expression Omnibus (accession no: GSE69168) |

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
