## [Decision Letter]

Thank you for submitting your article "Mapping the mouse Allelome reveals tissue-specific regulation of allelic expression" for consideration by *eLife*. Your article has been reviewed by two peer reviewers, and the evaluation has been overseen by a Reviewing Editor and Fiona Watt as the Senior Editor. The reviewers have opted to remain anonymous.

The reviewers have discussed the reviews with one another and the Reviewing Editor has drafted this decision to help you prepare a revised submission.

We have sent your manuscript out to two reviewers for their evaluations. Below you will find the reviews provided by these reviewers. As you will see, each of the reviewers agrees that there are potentially important biological insights contained in this manuscript. However, there are several issues relating to clarity and the presentation of the data described. While at this moment we conclude that your manuscript is not ready for publication in *eLife*, a revised version that addresses the issues raised by the reviewers would be of interest.

Summary:

This manuscript entitled "Mapping the mouse Allelome reveals tissue-specific regulation of allelic expression" Andergassen et al. present a comprehensive analysis of the mouse Allelome. The authors generate RNA-Seq data from 23 different developmental stages including 19 female tissues from F1 crosses between FVB and CAST mice. The results of the authors' analyses show that allele specific expression is highly tissue-specific and also show that this behavior is regulated by tissue-specific enhancers. In so doing, they identify imprinted genes and compare against the current literature. Lastly, they show that the allelic behavior occurs in larger genomic clusters than previously expected.

While the data presented in this manuscript could be an interesting and useful resource describing allelic behavior in the mouse genome there are a number of concerns (listed below) that should be addressed to provide some clarity to the conclusions and to assist readers in their comprehension.

Essential revisions:

1) The authors use the own software Allelome. PRO that has previously been published in NAR to perform their allelic analysis. They are not clear about the parameters and thresholds that they use to perform these analyses. In Figure 1—figure supplement 1 they present the "allelic score cutoff" which is never defined. They use an allelic ratio cutoff of 0.7 (they use a ratio of 0.6 for the XCI analysis on the basis that it is more stringent) which seems somewhat arbitrary and never justify its choice (or the dependence of their results on these parameters).

2) Their definition of bi-allelic expression (BAE) includes genes that have allelic expression ratios of less than 0.7 but also includes genes that are greater than 0.7 but are inconsistent between replicates – shouldn't this second category of genes be called ambiguous rather than biallelic and separated out?

3) Near the start of the paper (Results section) the authors classify genes as strain-biased for either CAST of FVB – these are a different set of genes from the maternal and paternal allele-expressed genes that they also find. They use these sets of genes throughout the manuscript. In looking throughout the paper including the supplemental methods there appears to be no definition as to what the authors mean by strain-biased genes (or define the parameters used to call genes in this category).

4) In their analysis of regulation of allele-specific genes the authors use the H3K27ac signal in a window of +/- 50 kb from the TSS of the target gene. They then use a procedure of computing the enrichment of allelic behavior of H3K27ac windows of 4kb in size compared to randomly shuffled windows. First, H3K27ac signal is localized to the TSS of an active gene, thus why is such a large window around the TSS needed – shouldn't it be localized no more than a kb from the TSS. Additionally, why perform the enrichment compared to a shuffled null? Can't one just compute the allelic signal of a genomic window centered on the TSS and compare the direction of the allelic bias (if there is one) with the target gene being regulated?

More technical issues:

1) In subsection “Escape from X-inactivation is tissue-specific and correlates with increased distance from monoallelic enhancers” they compute the significance of the escapers being further from H3K27ac windows compared to non-escapers and quote a significance of p<1e-20 and refer to the material and methods. In the Material and methods however, the authors present the analysis for this claim and show that the significance using a Fisher's exact test is p<1e-17.

2) In both Figure 3 and Figure 2—figure supplement 2C the use of similar colors such as green, blue and similar shades to correctly distinguish the features of the figures.

3) While "Allelome" seems OK, the use of "Escapome" seems unnecessary and unhelpful.

---

## [Author Response]

*Essential revisions:*

*1) The authors use the own software Allelome. PRO that has previously been published in NAR to perform their allelic analysis. They are not clear about the parameters and thresholds that they use to perform these analyses. In Figure 1—figure supplement 1 they present the "allelic score cutoff" which is never defined. They use an allelic ratio cutoff of 0.7 (they use a ratio of 0.6 for the XCI analysis on the basis that it is more stringent) which seems somewhat arbitrary and never justify its choice (or the dependence of their results on these parameters).*

In this study we largely use the Allelome.PRO parameters that we established in the Andergassen et al., 2015 Nucleic Acids Res paper. There we showed that most known imprinted genes had an allelic ratio cutoff above 0.7, a cutoff that also captured 85% of genes showing a known strain bias in expression on the X chromosome between CAST and FVB. We have now inserted a sentence into the Results section of this paper emphasizing this point to explain why we chose the 0.7 allelic ratio cutoff (subsection “The mouse gene expression Allelome shows tissue-specific variation”). The Allelome.PRO parameters and thresholds used for most analyses are given in the main text in the Results section, while the parameters for each specific analysis are given in the figure legends and Material and methods.

The allelic score is the log_10_(p) probability value that the allelic ratio deviates from 0.5 calculated based on the binomial distribution, with the direction of bias indicated by an assigned positive or negative value (imprinted bias: Maternal +ve, Paternal –ve; strain bias: CAST +ve, FVB -ve). The allelic score cutoff was determined by the false discovery rate (FDR), which was set at 1% for all analysis in this study. That is, the allelic score cutoff was set at the level where the number of genes passing the cutoff in mock comparisons was 1% of that observed in the true comparisons, as described in detail in the Nucleic Acids Res paper. We now introduce the allelic score cutoff in the main text (subsection “The mouse gene expression Allelome shows tissue-specific variation”).), and describe in detail how it is defined in the Material and methods (subsection “Allelome.PRO analysis of RNA and ChIP-seq data”). In addition we have modified Figure 1—figure supplement 2 to make it clearer to the readers how the Allelome.PRO pipeline works.

We classified X-linked genes as X chromosome inactivation (XCI) escapers if they did not show the expected CAST strain bias due to skewed XCI where the FVB allele was more frequently inactivated. In most cases XCI escapers were biallelic, and since the mean allelic ratio of X-linked genes was around 0.7 (0.735 in MEFs), many genes showing a consistent CAST bias would be classified as biallelic with a 0.7 allelic ratio cutoff (Andergassen et al., 2015). Therefore, as the reviewer notes, we chose a 0.6 allelic ratio cutoff for the XCI escaper analysis to be more conservative in classifying genes as biallelic and XCI escapers (subsection “Escape from X-inactivation is tissue-specific and correlates with increased distance from monoallelic enhancers”).

*2) Their definition of bi-allelic expression (BAE) includes genes that have allelic expression ratios of less than 0.7 but also includes genes that are greater than 0.7 but are inconsistent between replicates – shouldn't this second category of genes be called ambiguous rather than biallelic and separated out?*

In response to this comment we have analyzed the composition of biallelic genes, the results of which are shown in the new sub-figure Figure 1—figure supplement 2 and referenced in the Results section (subsection “The mouse gene expression Allelome shows tissue-specific variation”). We found that depending on the tissue, between 40.4-57.0% biallelic genes showed a consistent strain bias where the median ratio of the replicates was below the allelic ratio cutoff. The remaining biallelic genes, which included genes with an imprinted bias below the allelic ratio cutoff, showed the direction of strain bias fluctuated between the replicates. Of these a minority, between 4.8-10.4% of total biallelic genes, showed an inconsistent bias with at least one replicate over the 0.7 allelic ratio cutoff. Biallelic genes are negatively defined as expressed genes that do not show a consistent strain bias or imprinted bias over the allelic ratio and allelic score cutoffs, therefore we believe it is appropriate to retain these genes in the biallelic category rather than creating an extra ambiguous category.

*3) Near the start of the paper (Results section) the authors classify genes as strain-biased for either CAST of FVB – these are a different set of genes from the maternal and paternal allele-expressed genes that they also find. They use these sets of genes throughout the manuscript. In looking throughout the paper including the supplemental methods there appears to be no definition as to what the authors mean by strain-biased genes (or define the parameters used to call genes in this category).*

Strain biased genes are those genes that show a consistent expression bias towards either the FVB or CAST allele in all replicates above the allelic score FDR cutoff and with a median allelic ratio above 0.7. This differs from genes that show an imprinted bias, where the bias towards the FVB or CAST allele switches depending on the strain of the parents and whether the gene shows a maternal or paternal bias. To further clarify the classification of strain biased genes we have expanded the explanation of the Allelome.PRO pipeline in the Material and methods (subsection “Allelome.PRO analysis of RNA and ChIP-seq data”) and modified Figure 1—figure supplement 2 to show how each of the allelic categories are defined.

*4) In their analysis of regulation of allele-specific genes the authors use the H3K27ac signal in a window of +/- 50 kb from the TSS of the target gene. They then use a procedure of computing the enrichment of allelic behavior of H3K27ac windows of 4kb in size compared to randomly shuffled windows. First, H3K27ac signal is localized to the TSS of an active gene, thus why is such a large window around the TSS needed – shouldn't it be localized no more than a kb from the TSS. Additionally, why perform the enrichment compared to a shuffled null? Can't one just compute the allelic signal of a genomic window centered on the TSS and compare the direction of the allelic bias (if there is one) with the target gene being regulated?*

H3K27ac is enriched on active promoters and enhancers. The reviewer is correct that the H3K27ac bias around the TSS should follow the expression bias observed in RNA-seq, but in the analysis referred to here from Figure 2, we aimed to detect allelic biases in enhancers that may regulate strain biased expression. We chose a +/-50kb window around the TSS, because most enhancers regulate genes within this range (Sanyal et al., 2012), and excluded the region +/- 2kb from the TSS to remove promoter enriched H3K27ac from the analysis. It was necessary to compare the enrichment to shuffled windows to show that the associate that we saw was not random.

*More technical issues:*

*1) In subsection “Escape from X-inactivation is tissue-specific and correlates with increased distance from monoallelic enhancers” they compute the significance of the escapers being further from H3K27ac windows compared to non-escapers and quote a significance of p<1e-20 and refer to the Material and methods. In the Material and methods however, the authors present the analysis for this claim and show that the significance using a Fisher's exact test is p<1e-17.*

The main text is now corrected to give the correct value, p<10^-17^ (paragraph three, subsection “Escape from X-inactivation is tissue-specific and correlates with increased distance from monoallelic enhancers”).

*2) In both Figure 3 and Figure 2—figure supplement 2C the use of similar colors such as green, blue and similar shades to correctly distinguish the features of the figures.*

For most figures we used the color-code introduced in Figure 1 to indicate the different allelic states that we classify, taking care to chose colors that could be easily distinguished from each other. In Figure 3 to highlight XCI escapers on the Circos plot we made non-escaper genes partially transparent (CAST in embryonic and adult tissues, MAT in extra-embryonic tissues, we now state this in the figure legend). In Figure 2—figure supplement 1 we show a heatmap with a gradient of color from strain biased CAST (brown) to biallelic (green) to FVB strain biased (turquoise), allowing the degree of allelic bias to be visualized. For both these figures we believe the variation on our standard color-code improves understanding of the data, and therefore would prefer to keep the color-code in these figures as they are.

*3) While "Allelome" seems OK, the use of "Escapome" seems unnecessary and unhelpful.*

We have replaced the term “Escapome” in the text.